# Prompt Engineering Techniques for Language Model Reasoning Lack Replicability

**Laurène Vaugrante**                  *laurene.vaugrante@iris.uni-stuttgart.de*
*Interchange Forum for Reflecting on Intelligent Systems*
*University of Stuttgart*

**Mathias Niepert**
*Institute for Artificial Intelligence*
*University of Stuttgart*

**Thilo Hagendorff**
*Interchange Forum for Reflecting on Intelligent Systems*
*University of Stuttgart*

**Reviewed on OpenReview:** *https://openreview.net/forum?id=bgjR5bM44u*

## Abstract

As large language models (LLMs) are integrated into everyday applications, research into prompt engineering techniques (PET) to improve these models' behavior has surged. However, clear methodological guidelines for evaluating these techniques are lacking. This raises concerns about the replicability and generalizability of the prompt engineering techniques' benefits. We support our concerns with a series of replication experiments focused on zero-shot prompt engineering techniques purported to influence reasoning abilities in LLMs. We tested GPT-3.5, GPT-4o, Gemini 1.5 Pro, Claude 3 Opus, Llama 3, Vicuna, and BLOOM on the chain-of-thought, Sandbagging, EmotionPrompting, Re-Reading, Rephrase-and-Respond (RaR), and ExpertPrompting prompt engineering techniques. We applied them on manually double-checked subsets of reasoning benchmarks including CommonsenseQA, CRT, NumGLUE, ScienceQA, and StrategyQA. Our findings reveal a general lack of statistically significant differences across nearly all techniques tested, highlighting, among others, several methodological weaknesses in previous research. To counter these issues, we propose recommendations for establishing sound benchmarks, and designing rigorous experimental frameworks to ensure accurate and reliable assessments of model outputs.

## 1 Introduction

The field of generative artificial intelligence has considerably evolved in only a few years. In particular, large language models (LLMs) have witnessed an unprecedented surge in popularity with the release of ChatGPT (OpenAI, 2022), which became the most rapidly adopted internet application in history. LLMs possess advanced natural language processing capabilities which demonstrate a broad range of downstream applications, ranging from casual conversations to complex problem-solving (Minaee et al., 2024; Zhou et al., 2020). Given the fast growing range of applications (Guo et al., 2024) plus their respective risks for AI alignment (Ji et al., 2024), fairness (Hao et al., 2023), and safety (Amodei et al., 2016; Hagendorff, 2024; Vaugrante et al., 2025; Weidinger et al., 2023), it is paramount to evaluate behavioral and reasoning patterns these models exhibit (Binz & Schulz, 2023; Gao et al., 2025; Wang et al., 2024a). This created the need for new research fields, and fostered a variety of different approaches to investigate different types of LLM behavior, including emergent abilities and prompt engineering strategies (Chang et al., 2023). A substantial part of this research relies on prompt engineering techniques, which are designed to steer LLMs toward desired responses without modifying their internal structure (Liu et al., 2021). However, prior work has

demonstrated that some of these techniques fail to replicate in newer, reasoning-oriented models, particularly in software-engineering contexts (Wang et al., 2024b). We hypothesize that such replication challenges may extend beyond reasoning models and occur even with instruction-following LLMs across a broader range of reasoning tasks, indicating a deeper replication problem within prompt-engineering research. To investigate this, we conduct experiments attempting to conceptually replicate studies investigating zero-shot prompt engineering techniques that are believed to enhance reasoning in LLMs. Our findings reveal that these techniques often fail to produce consistent improvements. Based on our findings, we propose specific recommendations and tools for developing better methodologies when evaluating LLM behavior. This includes establishing sound benchmarks, designing robust experimental frameworks adapted to the LLMs used, and implementing accurate evaluations of model outputs.

## 2 Methods

### 2.1 Prompt Engineering Techniques

For our experiments, we tried to replicate single-pass, zero-shot prompt engineering techniques that were demonstrated to alter reasoning performances in LLMs in previous studies:

- **Zero-shot chain-of-thought Prompting** (Kojima et al., 2022): This method claims that adopting a step-by-step reasoning approach in LLMs enhances overall reasoning accuracy.

- **Sandbagging** (Perez et al., 2022): Sandbagging showcases that LLMs have a tendency to repeat back a dialog user's preferred response and mirror them when solving tasks.

- **EmotionPrompting** (Li et al., 2023): This technique consists in adding emotional stimuli, such as "This is very important to my career", in order to enhance the performance.

- **Re-Reading** (Xu et al., 2024): This method consists in repeating the task twice to enhance the reasoning accuracy.

- **Rephrase-and-Respond** (Deng et al., 2024): This approach involves rephrasing the given task as a query before providing a response, thereby improving accuracy.

- **ExpertPrompting** (Xu et al., 2025): This technique claims to enhance the LLM accuracy when setting the LLM in an expert role.

With this selection, we aim at covering the majority of prompt engineering techniques that represent zero-shot, single-pass methods which are most likely to be adopted by a wide range of users in real-world LLM settings. Furthermore, these techniques simplify the experimental design and minimize potential sources of error, which would be present in more complex settings. We deliberately excluded methods such as ensembling, self-criticism, least-to-most prompting, tree-of-thoughts, etc., as these involve more complex setups or prompt chains (Schulhoff et al., 2024; Yao et al., 2023) which we consider unlikely to be utilized by typical LLM users. As prompt engineering techniques can be applied to various applications (Liu et al., 2021) like improving text quality, enhancing the answer's relevance or controlling the output formatting, we chose to focus on prompt engineering techniques used to enhance reasoning abilities and hence accuracy of LLMs when prompted with complex tasks.

### 2.2 Benchmark Selection

To replicate the claimed impact of the selected prompt engineering techniques on LLM reasoning abilities, we selected five different benchmarks, each measuring a different type of reasoning: CommonsenseQA (Talmor et al., 2019), CRT (Hagendorff et al., 2023), NumGLUE (Mishra et al., 2022), ScienceQA (Lu et al., 2022) and StrategyQA (Geva et al., 2021). In accordance with a growing body of research (Gema et al., 2024; Goetze & Abramson, 2021) we noticed a low quality of many benchmark items, meaning incorrect or ambiguous questions, formatting flaws, or factual errors in the response choices. Therefore, we chose to hand-pick

(through rule-based filtering and manual checks) 150 faultless questions out of a random sample of 200 questions per benchmark, with a total of $n = 750$, preferring accuracy over large sample sizes. The tasks were either open-ended, Boolean, or multiple-choice questions.

## 2.3 Experiments

We first measured the accuracy of LLMs in a base test using unmodified tasks. We then applied the prompt engineering techniques outlined in the studies mentioned above by incorporating the necessary pre- or suffixes to each task. We used the same prompts described in these studies when available, and generated new ones based on the prompt descriptions when they were not. When the studies used several pre- or suffixes as a basis to their claim, such as in the EmotionPrompting study where 11 different emotional stimuli were used, we randomly selected one of them for each task using a seed.

## 2.4 LLM selection

We compared the performance of five different LLMs, in particular OpenAI's GPT-3.5 (gpt-3.5-turbo ) (OpenAI, 2022) and GPT-4o (gpt-4o-2024-05-13) (OpenAI, 2023), Google's Gemini 1.5 Pro (gemini-1.5-pro-001) (Team et al., 2024), Anthropic's Claude 3 Opus (claude-3-opus-20240229) (Anthropic, 2024), Google's Gemini 1.5 Pro (gemini-1.5-pro-001) (Team et al., 2024), and Meta's Llama 3, with both 8B and 70B versions (Meta-Llama-3-8B-Instruct and Meta-Llama-3-70B-Instruct) (34). As the selected studies also used some models from earlier generations, we have also attempted to measure the performance for Vicuna 13B v1.5 (Chiang et al., 2023) as well as BLOOM 176B (Workshop et al., 2023), but the obtained results were deemed unusable due to the models' inability to generate coherent outputs, their tendency to produce meaningless loops, repeated fragments of the input, and other issues, as detailed in Appendix A.

## 2.5 Output classification

To facilitate the LLM output classification process without restricting the reasoning behavior during the LLMs' prompt completions, we added an instruction to write the final answer after a specific string, namely "####", to each benchmark task, as indicated in the literature (Cobbe et al., 2021; Nezhurina et al., 2024). We then assessed the LLM outputs following "####" by combining string matching methods, LLM-based evaluations with GPT-4o, as well as manual double-checks (see Appendix B). Considering that the behavior of LLMs might exhibit variations over time (Chen et al., 2024a), we report the timeframe of the experiments. They spanned from June 6th, 2024, to June 17th, 2024, except for the Vicuna 13B and BLOOM experiments, which spanned from December 4th to December 14th, 2024. For all experiments, LLM temperature parameters were set to 0, or 0.00001 when 0 was not permitted.

## 2.6 Study focus

This study focuses on *replication* rather than on *reproducibility*. According to Peng (2011), replication involves collecting and analyzing new data to replicate the findings of a previously conducted study, whereas reproducibility entails reanalyzing the original data to verify the results. Our hypothesis when replicating the previous experiments was that the claimed performance improvements are not replicable and hence the claims about the prompt engineering techniques are not *generalizable*. We neither use the exact same selections of benchmarks nor models as in the original studies but vary the experimental setups slightly. In detail, this means that we still test foundation text-to-text models, use reasoning benchmarks, and use prompt engineering techniques either verbatim (chain-of–thought, Re-Reading, EmotionPrompting, Rephrase-and-Respond), or, if necessary, adapted in alignment with their original methodology to suit our benchmarks (Sandbagging, ExpertPrompting,), which should theoretically increase their utility. However, it is important to note that our approach to classifying LLM responses likely differs substantially from the original studies, as many of them lack descriptions of their chosen methods, further underscoring the nature of our work as a replication study rather than a reproduction.

## 2.7 Statistics

All statistical analyses were performed using Python (version 3.11.4). The SciPy library (version 1.13.1) was used for statistical computations, while visualizations were created with Matplotlib (version 3.7.1) and Seaborn (version 0.12.2). We applied chi-squared ($\chi^2$) tests to assess the statistical significance of accuracy differences between baseline and modified prompts. Rounded P-values are reported for each test. 95% confidence intervals (CIs) were calculated and included in the result visualizations. Performance variability across models and benchmarks was accounted for, and results were reported per model, benchmark, and prompt engineering technique.

# 3 Results

## 3.1 Chain-of-thought prompting

Chain-of-thought prompting involves decomposing a given task and solving each step before outputting the final answer, by presenting the LLM with an example of a task and its expected decomposed output. In the original study establishing this method, Wei et al. (2023) tested five LLMs over three reasoning, categories including arithmetic reasoning, commonsense reasoning, and symbolic reasoning, harnessing 12 different benchmarks. The authors claim a good robustness of this method, with several different annotators. While they reported variance in the average performance, it was consistently superior to the performance with the base evaluation, with a reported average improvement of 39.91% (Wei et al., 2023). A subsequent study then claimed that a zero-shot chain-of-thought prompting strategy sufficed to elicit similar improvements (Kojima et al., 2022). Instead of presenting, before each task, an example enabling chain-of-thought reasoning, they simply suffix tasks with "Let's think step by step". They tested a larger sample of 17 LLMs on various reasoning categories, utilizing 12 benchmarks akin to the previous paper. They obtained an averaged 35.93% improvement in accuracy for zero-shot chain-of-thought reasoning across all benchmarks and models (Kojima et al., 2022). We tried to replicate these findings with our set of reasoning benchmarks. However, despite the impressive results from the original studies, we observed that there was no significant improvement (see Figure 1): with the exact same task suffix as in the original study, we could not observe any significant difference across all benchmarks. With results from all models combined, the maximal positive impact of chain-of-thought reasoning is with NumGLUE where there is a 2.78% accuracy difference between the base and the chain-of-thought prompt (see Appendix C), which is not significant given the total number of tasks ($\chi^2 = 1.78, p = 0.18$). These numbers remain similar throughout each LLM evaluated, with an overall average improvement of 0% for the chain-of-thought reasoning ($\chi^2 = 0.06, p = 0.8$), as seen in Appendix C. The largest observed positive impact of chain-of-thought reasoning is for Llama 3-70B tasked by CommonsenseQA, with an observed 8.67% improvement ($\chi^2 = 2.19, p = 0.14$) (see Appendix C). However, the highest overall difference is an 11.33% accuracy decrease ($\chi^2 = 4.47, p < .05$) (see Appendix C) with chain-of-thought reasoning applied on Llama 3-70B with StrategyQA. While the latest models seem to implement chain-of-thought reasoning by default, meaning without being specifically prompted to, these results hold even for previous models such as GPT-3.5, which often do not. We compared the average response length of each LLM when chain-of-thought reasoning is explicitly requested, compared to when it is not, as shown in Appendix D. Even when the base experiments do not demonstrate verbose prompt completions and the chain-of-thought prompting does, the performance results are not impacted in a significant manner, which stands contrary to what the literature suggests (Jin et al., 2024). For instance, GPT-4o had an average difference of response lengths of 531 characters for the base test vs. 931 characters for the chain-of-thought prompting, but just a 0.01% accuracy difference, suggesting that simply increasing the length of prompt completions does not enhance accuracy beyond a certain point.

## 3.2 Sandbagging

Perez et al. (2022) demonstrate sycophancy, which is an LLM's tendency to output answers that users tend to prefer. The researchers evaluated several aspects of sycophancy, including a "sandbagging" capability, which suggests that a model could underperform when a user is deemed incapable to solve or verify a given task. They underpin this hypothesis by adding user biographies before reasoning tasks from TruthfulQA (Lin et al.,

2022), with "very educated" users as opposed to "very uneducated" users. They imply a significant difference between these two categories, claiming that sandbagging causes LLMs to output incorrect answers when human users are perceived as unable to answer correctly themselves (Perez et al., 2022). We conceptually replicate this experiment using our selected models by prefixing our selected reasoning tasks with both "very educated" and "very uneducated" user biographies (see Appendix E). We observe no significant difference over all benchmarks when comparing the highly educated ($\chi^2 = 1.64, p = 0.20$) or poorly educated ($\chi^2 = 1.24, p = 0.27$) user prompts to the base results (see Figure 1 and Appendix C), with an average accuracy decrease of 1% for both cases (see Appendix C). We likewise observe no significant difference when comparing the highly educated to the poorly educated user prompt results, and frequently observe that the "poor education" prefixed tasks have an even better performance than the "high education" ones (average accuracy improvement of 0.1% for "poor education"). Once again, we fail to replicate the sandbagging phenomenon when utilizing our experimental setup.

### 3.3 EmotionPrompting

EmotionPrompting, presented by Li et al. (2023), augments a task with emotional cues such as "You'd better be sure" or "This is very important to my career" to enhance problem-solving abilities in LLMs. In the original study, Li et al. (2023) augmented tasks with 11 variations of emotional stimuli and tested six LLMs including ChatGPT and GPT-4. They sourced their tasks from three different benchmark categories, notably using tasks from BIG-Bench (Srivastava et al., 2022). They claim to obtain a "relative performance improvement of 115%" (Li et al., 2023) with their method, arguing that adding an emotional component improves the capabilities of LLMs. However, despite the improvement that was strongly implied throughout the original study by raising claims like "EmotionPrompt makes it easy to boost the performance of LLMs" (Li et al., 2023), the numerical values communicated in the study itself do not coincide with these claims. Instead of communicating the average improvement of the enhanced prompts over the regular prompts, they focused on improvements when cherry-picking the most performant emotional cue. Based on their reported results, we calculated an average relative performance improvement of 4.42% on BIG-Bench tasks, and a 2.58% relative performance improvement across all benchmarks, when choosing the average performance of all emotional stimuli. Despite identifying this shortcoming in the original study at this early stage, we nevertheless replicated the experiments with our selected tasks and models. We applied the same emotional suffixes as in the original study, apart from "Are you sure?", as LLMs tend to reply to this question, as opposed to solving the given tasks. Similarly to Li et al. (2023)'s findings, but contrary to their claims, we observed that there was no significant improvement, across every single model and benchmark (see Figure 1). The maximal positive improvement measured is non-significant with an 8.7% difference ($\chi^2 = 1.94, p = 0.16$) (see Appendix C), using Llama 3-8B on CommonsenseQA. Overall, we observe an insignificant performance increase of 1% when applying EmotionPrompting ($\chi^2 = 0.11, p = 0.74$) (see Appendix C).

### 3.4 Re-Reading

Re-Reading, introduced by Xu et al. (2024), consists in repeating the task verbatim before having the model answer. They compared the baseline performance with the Re-Reading performance, as well as the performance in both conditions when additionally suffixing every task with a chain-of-thought eliciting prompt. The researchers tested GPT-3 (text-davinci-003) (Brown et al., 2020), GPT-3.5, Llama-2-13B and Llama-2-70B (Touvron et al., 2023), in order to have both models with and without instruction fine-tuning. They used a total of 112 arithmetic, common sense, and symbolic reasoning tasks sourced from various datasets with GPT-3 and GPT-3.5, for which they obtained an average gain of 2.7% in accuracy, and 2.9% with the inclusion of chain-of-thought reasoning. For Llama 2-13B and Llama 2-70B, they used a different set of benchmarks comprising only arithmetic reasoning tasks, with an average gain of 2.5% in accuracy (2.7% with chain-of-thought reasoning). We replicate the Re-Reading experiments on our selected tasks and models. For this study, we observe a significant improvement for Llama 3-8B ($\chi^2 = 13.13, p < .05$) and Llama 3-70B ($\chi^2 = 19.4, p < .05$) exclusively (see Appendix C and Figure 1). The maximal improvement across all benchmarks for the other models is of 2%, for Claude 3 Opus ($\chi^2 = 1.27, p = 0.26$). Therefore, Re-Reading seems replicable for the Llama 3 models only, which highlights the importance of implementing tests on a variety of models. However, the initial study indicated that Re-Reading was effective on GPT

models, notably GPT-3.5, that we also tested with different outcomes. Therefore, we only managed to partially replicate the results.

### 3.5 Rephrase-and-Respond

Rephrase-and-Respond, inspired by the communication technique of rephrasing to enhance clarity, involves instructing an LLM to first rephrase a task and then address it within the same prompt. Deng et al. (2024) demonstrate that simply adding a directive such as "Rephrase and expand the question, and respond" can significantly enhance LLM accuracy. They evaluated GPT-3.5, GPT-4 and Llama 2 across seven different reasoning benchmarks, such as CommonsenseQA and Knowledge Classification (Allen-Zhu & Li, 2024), and observed an average improvement of 17.80% across all tested models. We replicated this experiment by incorporating the suggested rephrasing directive to reformulate questions across our full dataset. We then applied a tailored algorithm to classify the answers, ensuring it could accommodate these new types of responses (see Appendix B). As for the Re-Reading experiment, we only observe a significant positive effect for Llama 3-8B ($\chi^2 = 10.49, p < .05$) and Llama 3-70B ($\chi^2 = 18.63, p < .05$), with the StrategyQA benchmark. Once again, Rephrase-and-Respond seems to only show a significant positive improvement on a specific benchmark-model combination, which shows only a partial replication of the results (see Figure 1 and Appendix C).

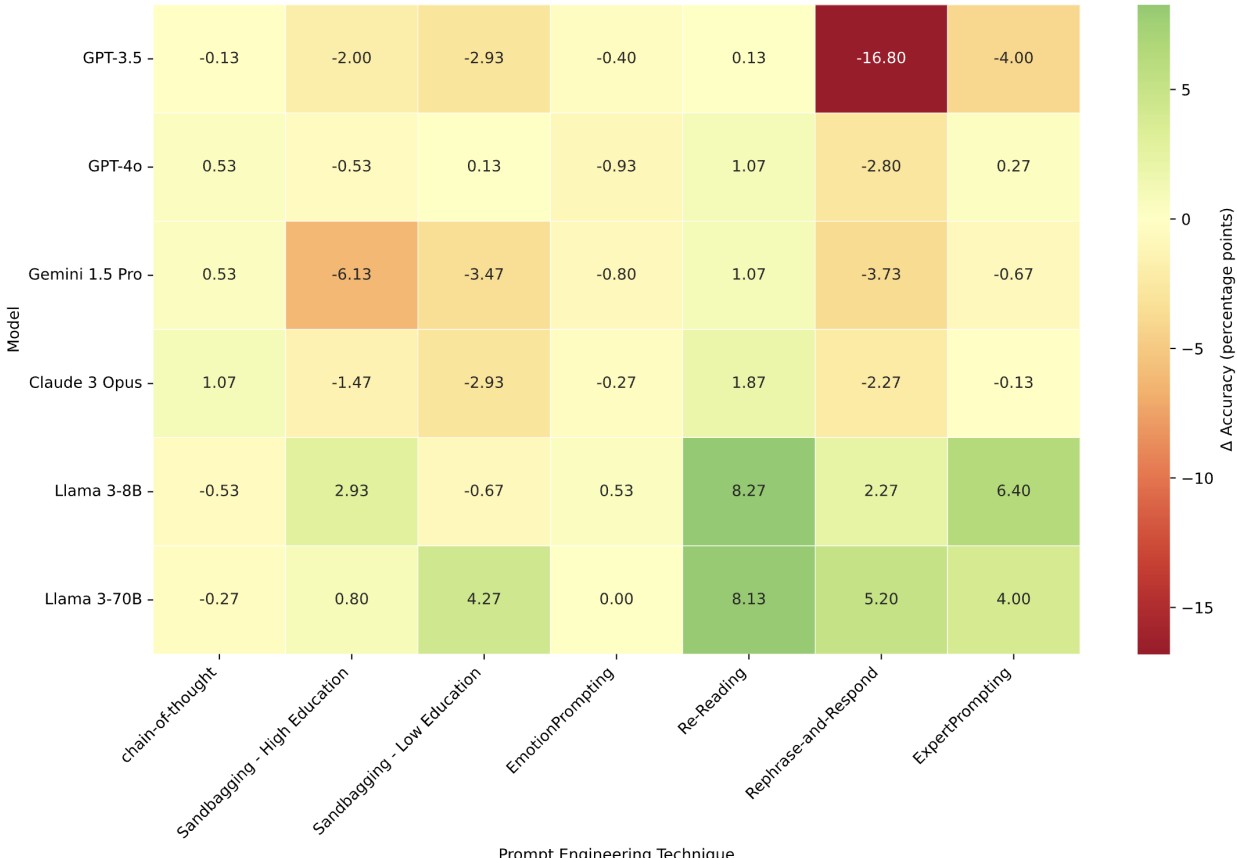

Figure 1: Accuracy comparisons between the base tests without any prompt modification and the augmented prompts across all LLMs.

### 3.6 ExpertPrompting

ExpertPrompting consists of giving LLMs an instruction to impersonate someone with high expertise on the task subject while completing a task. This method presented by Xu et al. (2025) has been greatly

popularized; it is now recommended in LLM documentations for enhanced LLM accuracy and improved focus on adhering to the task's requirement. Xu et al. (2025) evaluated the response quality of ExpertPrompting, assessing aspects like accuracy, helpfulness, or relevance. They evaluated GPT-4 responses with and without ExpertPrompting, which, in the case of the former, possessed a reported higher answer quality 48.5% of the time (Xu et al., 2025). In our experiments, we measure, as with the previous prompt engineering techniques, the accuracy of the ExpertPrompting technique using our set of reasoning benchmarks and LLMs. We observe no significant improvement across all benchmarks ($\chi^2 = 1.57, p = 0.21$) (see Figure 1 and Appendix C), with an average improvement of only 1% (see Appendix C).

## 4 Identified Issues and Recommendations

Given the identified lack of replicability across various studies, we deem crucial to address the underlying issues contributing to these replication problems in prompt engineering. We have identified specific problems associated with each cited prompt engineering method, which may hint at a broader replication problem in LLM behavior research. In the following, we provide recommendations aimed at mitigating replication problems in prompt engineering and related fields, based on our analysis of the selected studies: (1) benchmark adequacy, (2) model variability and model-benchmark compatibility, and (3) output evaluation and reliability. It is important to note that while an absolute elimination of replication problems is unlikely, our recommendations aim to minimize such challenges effectively.

### 4.1 Benchmark adequacy

### 4.1.1 Benchmark Quality

In several of the studies examined, we noted major issues regarding the benchmarks used to assess LLM performances. Many benchmark tasks are flawed: they lack proper grammar, contain spelling or punctuation errors (e.g., missing question marks in multiple-choice questions), are semantically nonsensical, incomplete, or factually incorrect. For instance, for CommonsenseQA (used in the chain-of-thought, Re-Reading, and Rephrase-and-Respond studies), 10.9% of questions presented punctuation issues easily verifiable with a simple code. Such flaws are not isolated incidents: they reflect a broader pattern of benchmark instability that has been widely discussed in the literature (Gema et al., 2024; Goetze & Abramson, 2021). Label errors can destabilize benchmark reliability, and even small proportions of mislabeled test data can invert model rankings: a phenomenon they call the capacity–performance paradox (Northcutt et al., 2021). This effect implies that benchmark noise may disproportionately penalize higher-capacity models, creating misleading impressions of their reasoning ability. Similarly, researchers have revealed significant ground-truth errors in the Massive Multitask Language Understanding (MMLU) benchmark, one of the most widely used evaluation suites for LLMs (Gema et al., 2024). Their re-annotated version, MMLU-Redux, shows that previously reported model hierarchies often vanish once test data are corrected. Likewise, prior work has exposed flaws in widely used reasoning benchmarks, revealing that evaluation setups often reward superficial lexical alignment rather than genuine reasoning (Mousavi et al., 2025). Together, these findings align with our own observations: large benchmark datasets frequently contain nonsensical, ambiguous, or incorrect items that can distort results.

To mitigate benchmark-induced replication failures, we recommend that researchers rigorously validate benchmark datasets before use. Automated scripts can detect obvious errors (e.g., typos, missing punctuation, malformed options), but human verification remains essential for identifying semantic inconsistencies. Following research recommendations (Northcutt et al., 2021; Cao et al., 2025), benchmark curation should be treated as a core methodological step rather than an auxiliary task. We recommend mentioning the cleaning procedure and error rates, and encourage the adoption of transparent guidelines such as How2Bench (Cao et al., 2025). Where possible, studies should prioritize quality over quantity, using smaller but validated subsets rather than relying on noisy large-scale test sets. This approach minimizes variance, prevents benchmark-driven illusions of improvement, and enhances comparability across studies. Finally, we advocate for the publication of "benchmark versioning" and reproducible evaluation code, so that future research can easily verify whether results stem from model performance or from artifacts in the evaluation data.

### 4.1.2 Benchmark Coherence and Cross-Study Comparability

A second major source of replication instability arises from inconsistent benchmark selection within and across studies. In several prompt-engineering papers, models are evaluated on distinct benchmark sets within the same experiment. For example, in the Re-Reading study (Xu et al., 2024), GPT models were tested on three reasoning benchmarks, whereas Llama 2 models were evaluated on only one. In the sandbagging study (Perez et al., 2022), researchers evaluate differences in model accuracy when answering questions on the TruthfulQA dataset (Lin et al., 2022), which measures whether a language model is truthful in generating answers to questions, so whether the facts mentioned in the answer are correct rather than assessing whether they answered the task correctly. Such inconsistency complicates direct comparison of model performance and raises questions about the rationale behind benchmark choices, potentially leading to benchmark cherry-picking that favors results supporting a given hypothesis. Beyond within-study inconsistencies, there is little coherence in benchmark use across related works. In our replication, we applied the same benchmark suite across all experiments to ensure comparability. In contrast, the six studies we examined drew on widely different datasets, some domain-restricted, others deterministic in nature. For instance, the chain-of-thought study evaluated reasoning-centric tasks, while the EmotionPrompting paper used tasks from Instruction Induction (Honovich et al., 2022) and BIG-Bench (Srivastava et al., 2022), such as rhyming or pluralization, which makes results hard to compare as these measure different underlying abilities. Similarly, variations within the process of administering benchmark tasks (notably zero-shot prompting versus few-shot prompting) impact the reasoning process of LLMs and therefore the outcomes. These issues mirror findings by researchers who conducted a large-scale analysis of benchmark agreement and showed that minor methodological differences, such as prompt format, evaluation metric, or dataset choice, can drastically alter perceived model performance (Perlitz et al., 2024). Finally, it is essential to select benchmarks that are conceptually and structurally aligned with the research question. Indeed, when testing prompt techniques claiming to alter reasoning performance, given or implied response instructions in the tasks can interfere with output accuracy by restricting the response length and therefore its ability to generate more detailed responses, for instance in multiple-choice settings.

Table 1: Benchmark-related replication issues, mitigation techniques, and tools

| Issue | Mitigation | Tools |
|---|---|---|
| **Flawed benchmark tasks**: may inflate or deflate perceived performance and favor some models | • Include benchmark curation as a core methodological step
• Prioritize quality over quantity
• Report quality-check methods and error rates | • Automated validation scripts
• Guidelines such as How2Bench (Cao et al., 2025) |
| **Inconsistent benchmark selection**: complicates cross-study comparison | • Report exact benchmarks and what they measure
• Adopt standardized benchmark-agreement protocols | • BenchBench framework (Perlitz et al., 2024)
• Benchmark agreement protocol |
| **Benchmark–objective misalignment**: may distort perceived performance by assessing unrelated capabilities | • Ensure coherence between benchmarks and research objectives
• Document rationale for benchmark choice | • BenchBench framework (Perlitz et al., 2024)
• Benchmark agreement protocol |

We recommend preferring standardized benchmarks across studies in the same field, to ensure that benchmarks are closely aligned with the research objectives. We would advise researchers to ensure that benchmark domains match the specific research objective. We further recommend adopting benchmark-agreement protocols such as those implemented in BenchBench (Perlitz et al., 2024), which promote standardized evaluation and transparent documentation of benchmark configurations. Establishing and reporting such consistency will reduce variance, mitigate cherry-picking concerns, and make replication efforts more meaningful.

### 4.2 Model Variability and Model-Benchmark Compatibility

### 4.2.1 Model Variability

In some cases such as in the chain-of-thought prompting study (Kojima et al., 2022), the lack of replicability is linked to the models' type and generation. Our results reveal a divergence between GPT-3.5 and GPT-4o: the former shows mild improvement with chain-of-thought prompting, whereas the latter displays no significant change. Although neither result reaches statistical significance, this difference suggests that for earlier-generation models (e.g., GPT-3), used in the replicated study along with other models of that generation, chain-of-thought prompting might have successfully improved LLM accuracy. This aligns with the system cards for recent models, which explicitly warn that techniques like chain-of-thought reasoning may not improve performance and can even impair it, advising caution in their use with these models (OpenAI, 2024). Similarly, we observed a significant improvement for some benchmarks with the Re-Reading and Rephrase-and-Respond prompts, for the Llama 3 models exclusively; if we look at the other models separately, the results are vastly non-significant (see Appendix C). This reinforces the claim that similar experiments may have a considerably different impact depending on the models used. Moreover, even when conducting replication experiments using the same models as in the original study, the opacity surrounding model updates and developer prompt modifications in terms of date and type of update (Chen et al., 2024a) renders study replications difficult. Furthermore, the efficacy of prompt engineering depends strongly on the model class; whether the model is inherently reasoning-oriented or relies on surface-level pattern completion. This dynamic is consistent with prior work demonstrating that prompt engineering substantially benefits non-reasoning LLMs but provides marginal or inconsistent improvements for reasoning-tuned ones such as o1-mini (Wang et al., 2024b).

Therefore, it is essential to use a variety of models when testing a hypothesis, or to at least mention the limited scope of the study when fewer models are used, as an effort to prevent a generalization that may be incorrect. In addition to that, some earlier-generation models may not be well-suited for such question-answering experiments requiring reasoning abilities. Specifically, we tested Vicuna 13B and BLOOM on our selected tasks (Appendix A), but found the results difficult to interpret, as the accuracy for both the base questions and the prompt-engineered questions was too low to draw reliable insights into the models' behavior. The studies on EmotionPrompting, Rephrase-and-Respond and ExpertPrompting used Vicuna or BLOOM nonetheless. We recommend consistently clarifying the methodology used for each model, as we were unable to understand how these studies achieved viable results with these models. We recommend that future studies:

1. Explicitly classify models by reasoning capacity when reporting results, distinguishing instruction-following models from reasoning-optimized ones.

2. Document model version and release date to ensure comparability over time, as minor provider-side updates can significantly alter performance.

Clarifying these methodological details will help disentangle true performance differences from artifacts introduced by model selection, and will prevent overgeneralization from results obtained on older, non-reasoning LLMs to newer architectures that already internalize these behaviors.

### 4.2.2 Model-benchmark Compatibility

As large language models evolve, benchmark saturation and data contamination have become central challenges in evaluating prompt-engineering methods. As newer models improve in baseline reasoning perfor-

mance, many existing benchmarks become effectively saturated, leaving little room for prompt engineering to demonstrate measurable gains. For example, tasks that once challenged early-generation models are often solved at near-ceiling accuracy by later models, as we can see with the CRT benchmark (Hagendorff et al., 2023). This saturation reduces statistical sensitivity and can obscure subtle effects introduced by prompting techniques. It is therefore essential to continuously recalibrate benchmark difficulty to match evolving model capabilities. Similarly, many zero-shot and few-shot benchmarks suffer from task leakage (Li & Flanigan, 2024; Wu et al., 2025; Zhang et al., 2025): models perform substantially better on datasets predating their training cutoff than on unseen ones, implying widespread contamination. These findings highlight that contamination can make both baseline and prompt-engineered performances appear inflated, thereby diminishing the apparent impact of prompting interventions.

Leakage-free benchmarks such as RandomCalculation (Wu et al., 2025) can be used to isolate genuine reasoning improvements. Likewise, researchers introduce a reasoning-driven synthesis framework that generates new research-level QA pairs automatically verified for correctness (Zhang et al., 2025). This method is effective for mitigating contamination and memorization effects. Moreover, we recommend using canary strings when publishing benchmark data to help detect and quantify potential data leakage in future models trained on that data. By embedding unique, traceable strings within benchmark datasets, researchers can later probe models to see whether these sequences are reproduced, thus revealing unintended memorization. This proactive approach further enhances the reliability of evaluation results.Finally, prompt effectiveness depends on both model architecture and task characteristics (Chen et al., 2024b). For an optimal performance, different prompts should be adapted to different LLMs, which may explain some differences observed in this study, especially when the original prompt engineering studies were applied on a small sample of LLMs. A "one-size-fits-all" benchmark evaluation approach can no longer capture the true impact of prompt-engineering methods, without comparing the exact same models. Other than expanding the number and variety of LLMs tested, a possible solution would be to adapt these prompt engineering techniques to each LLM using model-adaptive prompt optimizers (Chen et al., 2024b) to calibrate the prompts to the models.

We recommend that future studies:

1. Adjust benchmark difficulty to prevent saturation: when newer reasoning models reach near-ceiling performance, evaluations should shift toward more complex or reasoning-intensive tasks.

2. Mitigate contamination by verifying dataset release dates and removing items that predate model training cutoffs.

3. Employ model-adaptive evaluation strategies that align prompts and benchmarks with each model's capabilities in question, reducing misleading cross-model comparisons.

4. Report benchmark provenance and contamination checks as part of all experimental documentation, alongside model version and date, to promote transparency and interpretability.

5. Broaden model coverage to strengthen the generalizability of findings.

Ultimately, as LLMs integrate reasoning processes internally, older benchmarks cease to serve as meaningful discriminators of model capability or prompt-engineering effectiveness. Reliable evaluation now requires not only harder benchmarks such as Humanity's Last Exam (Phan et al., 2025), but also methodologies that explicitly account for model–benchmark interactions: how specific benchmarks align with, advantage, or contaminate particular models, thereby shaping observed results.

### 4.3 Output Evaluation and Reliability

### 4.3.1 Evaluation Reliability

For evaluations of LLMs, it is key to ensure the accuracy of the LLM output classifications. We have attempted to replicate a large number of verification techniques presented in other studies. However, when checking the accuracy of these techniques, we discovered that a significant number of them had shortcomings. For example, functions based solely on Regex rules were generally too vague, leading to flawed classifications.

Table 2: Model-related replication issues, mitigation techniques, and tools

| Issue | Mitigation | Relevant Tools |
|---|---|---|
| **Model variability and opacity**: performance differences across model generations and families | • Classify models by reasoning capacity (reasoning vs. non-reasoning)
• Document exact model version | |
| **Benchmark saturation and contamination**: newer models reach near-ceiling accuracy or show inflated results for pre-cutoff datasets | • Recalibrate benchmark difficulty using reasoning-intensive tasks
• Verify dataset release dates against model training cutoffs
• Use leakage-free datasets
• Check for canary strings | • Leakage-free datasets such as RandomCalculation (Wu et al., 2025)
• Beyond Memorization synthesis framework (Zhang et al., 2025)
• Harder benchmarks such as Humanity's Last Exam (Phan et al., 2025) |
| **Model–benchmark incompatibility**: some benchmarks advantage or disadvantage certain model types, skewing generalization | • Conduct cross-model evaluations on identical benchmarks | |

Other metrics, such as the F1 word overlap score, do not work effectively, as they would classify correct LLM outputs as incorrect, simply because the token length differed too much from the ground truth. Similar issues have been found with metrics like BLEU, ROUGE or METEOR (Ng & Abrecht, 2015; Liu et al., 2017). Moreover, studies often rely on using LLM-as-a-judge to classify LLM outputs (Pan et al., 2023), such as the ExpertPrompting study (Li et al., 2023). Exclusively relying on LLM-based evaluation can introduce flaws. When applying the original or even enhanced versions of their LLM output classification prompts, we observed multiple issues that compromised evaluation reliability. For instance, the judge would produce inconsistent evaluations when asked multiple times to assess the same output. It was also influenced by the task when classifying responses, and the verification prompt needed to be task-specific and highly precise. Prior research has also documented LLM-as-a-judge biases that may further distort classifications, such as the self-enhancement bias, a tendency for an LLM judge to rate outputs from its own model family more favorably, or the verbosity bias, where longer responses are rated more positively (Zheng et al., 2023). As some prompting techniques, such as chain-of-thought prompting, systematically increase response length (Appendix D), verbosity bias may artificially inflate measured accuracy when using LLM judges. Given the time-consuming aspect of manually double-checking classifications, we also suspect that this is done very rarely. Furthermore, among the papers we replicated, one did not include any details about the verification process (Li et al., 2023); it goes without saying that any verification process should be clearly reported in each study, to make the study replication feasible. We therefore recommend developing more precise and task-specific verification methods, and ensuring thorough documentation of these processes in all studies to facilitate accurate replication and validation of results. We recommend avoiding reliance solely on LLM-based evaluation; if alternative evaluation strategies are not feasible, then we would recommend running multiple iterations per prompt and aggregating the results to mitigate potential misjudgments. We also recommend that creators of new benchmarks provide a standardized verification process, encouraging all users to apply the same verification criteria.

### 4.3.2 Evaluation Metrics

The choice of evaluation metric is as critical as the verification method itself. Different metrics capture fundamentally distinct dimensions of model behavior, and results are not directly comparable across them. For instance, accuracy reflects factual correctness, whereas output quality captures fluency, coherence, or stylistic adequacy. Similarly, pairwise comparison approaches - where an LLM or human judge selects the "better" of two outputs - measure relative preference, not absolute performance. Treating these heterogeneous measures as equivalent can therefore produce misleading conclusions about the effectiveness of a prompting technique. For example, in our study, the original ExpertPrompting study focused on output quality rather than accuracy. We replicated their comparative evaluation approach using our own benchmarks. To ensure an unrestricted quality assessment, we removed formatting instructions from our prompts and evaluated complete LLM outputs. Our results showed higher quality scores for ExpertPrompting over base prompts, with winning percentages ranging from 26.53% ($\chi^2 = 82.65, p < .05$) for Gemini 1.5 Pro, to 76.93% ($\chi^2 = 294.27, p < .05$) for Llama 3–70B (see Figure 2). Despite these significant results, we remain skeptical about the robustness of this evaluation method. Relying solely on an LLM to assess outputs from another introduces risk of misjudgment, as seen in Section 4.3.1. Moreover, this pairwise evaluation method frequently selected a "winner" even when both responses were factually incorrect. In some cases, the model even favored an inaccurate response while penalizing an accurate one. Based on this observation, we re-evaluated the same responses for factual accuracy using a tailored version of our accuracy assessment. Once again, no significant improvement was observed. In fact, ExpertPrompting led to decreased accuracy, ranging from a drop of 2.4% ($\chi^2 = 0.84, p = 0.36$) for Gemini 1.5 Pro, to 14.53% ($\chi^2 = 34.00, p < .05$) for Llama 3–8B. This raises an important question: if a response is rated as "high quality" despite being factually incorrect, can it truly be considered an improvement? This can be misleading for users, who might reasonably assume that accuracy is embedded within the "quality" metric. Consequently, despite some studies presenting their claims similarly using verbs such as "improves", "enhances", "overperforms" to describe their prompting techniques, their outcomes cannot effectively be compared. Including both quality and accuracy metrics in evaluations would have provided a more comprehensive understanding of the effectiveness of, e.g., ExpertPrompting.

We would recommend including at least one factual metric when using reasoning benchmarks. Combining complementary metrics would also provide a more comprehensive picture of prompt effectiveness and guards against overinterpretation of single-score gains.

### 4.3.3 Reporting and Comparing Results Transparently

Across the replicated prompt-engineering studies, we observed substantial methodological variation that complicates the interpretation and comparison of results. Some studies notably display a particularly poor or unclear scientific method. In the EmotionPrompting study (Li et al., 2023), researchers cherry-pick the prompt with the best result out of eleven different prompts, rather than calculating an average across all prompts. This seemingly deliberate action may be due to a publication bias, which motivates researchers to manipulate results to be positive and therefore publishable. In addition, some studies, such as the Re-Reading study (Xu et al., 2024), report results as "significant" multiple times without presenting the corresponding statistical calculations or p-values. This lack of statistical transparency can mislead readers into assuming statistical significance without the necessary evidence to support such claims. It is crucial that when terms like "significant" are used, they are backed by clearly defined statistical measures. Moreover, some studies do not properly report the details of their experimental setup (Perez et al., 2022), which makes it confusing or even impossible to understand and therefore to replicate their process exactly. In this case, the lack of transparency forbids us from detecting possible shortcomings. We recommend adopting standardized evaluation methodologies and clearly defining metrics to ensure that results from different studies can be accurately compared and interpreted.

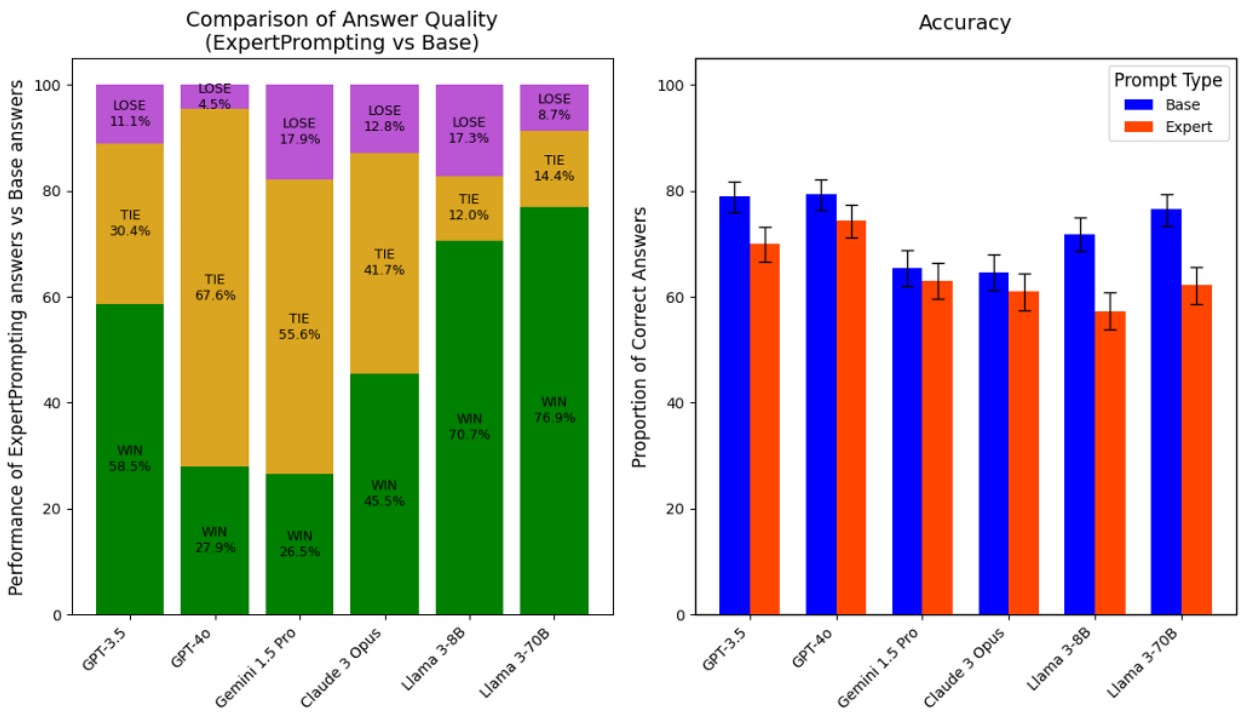

Figure 2: Comparison of accuracy and quality between the base tests and ExpertPrompting, using prompts without formatting constraints, across all LLMs. Error bars show 95% CIs.

Table 3: Results-related replication issues and mitigation techniques

| Issue | Mitigation |
|---|---|
| **Evaluation unreliability**: unstable or imprecise LLM-as-a-judge scoring | • Use precise, task-specific verification prompts
• Run multiple judging prompts and models and average across results
• Include manual or rule-based cross-checks |
| **Metric inconsistency**: mixing incomparable measures (e.g., accuracy vs. quality) | • Define all metrics clearly and report how each is computed
• Include at least one factual accuracy metric in reasoning tasks |
| **Cherry-picking and poor reporting**: selective results, missing p-values, or unclear methodology | • Aggregate across all tested prompts or justify subset selection
• Report all statistical tests, p-values, and confidence intervals
• Provide full methodological details (benchmarks, models, and prompts) |

## 5   Transparency Recommendations and Modular Evaluation Pipeline

To ensure methodological transparency and facilitate replication, future studies should explicitly document the following:

**Benchmarks**

- Exact benchmark names (and versions if applicable)

- Clear description of what each benchmark measures

- Quality-check methodology, if applied (e.g., automated validation scripts, manual review)

- Dataset provenance, including any filtering, sampling, or modification procedures

**Models**

- Exact model names and versions

- Description of model class (reasoning vs non-reasoning)

- Full inference methodology, e.g., temperature, max tokens, random seeds

- If used or modified, system prompts and/or added prompt instructions

**Evaluation and Metrics**

- Exact metrics reported (e.g., accuracy, quality, interpretability) and how they were computed

- If using LLM-as-a-judge: specify the judging model, version, full evaluation prompt, and aggregation method if any

- All statistical analyses: e.g., test type, p-values, confidence intervals

To support transparency and facilitate replication, we developed a modular evaluation pipeline that accompanies this study. The code was designed to allow researchers to replicate, extend, or adapt our experiments with minimal effort. Each stage of the evaluation process is modularized, enabling flexible modification of:

- **Benchmark selection:** users can easily substitute or add benchmarks, specify the number of random items drawn, and control sampling strategies.

- **Prompt-engineering techniques:** zero-shot prompting methods, particularly those relying on prefix or suffix additions, can be switched or customized through dedicated modules.

- **Model configuration:** the model used for inference can be easily changed, facilitating cross-model comparisons.

- **Evaluation and verification setup:** the output classification component allows users to define or modify the verification prompt and evaluate output quality or accuracy according to their own criteria.

This modular design serves a dual purpose. First, it allows rapid, small-scale experiments to obtain a preliminary estimate of the replicability of prior findings before committing to large-scale runs. Second, it encourages exploration and calibration: users can quickly test which combination of model, benchmark, prompting strategy, and evaluation prompt is most appropriate for their research context. The code is accessible here: `https://github.com/Laurene-v/replicatingPET`. By lowering the practical barrier to replication and promoting iterative experimentation, we aim to foster a culture of methodological transparency and empirical verification in prompt-engineering research.

## 6 Discussion

Our experiments show that most of the tested prompt engineering techniques do not lead to replicable or generalizable performance improvements in LLMs. Most techniques, when applied in slightly different experimental setups, failed to produce the claimed benefits. Some techniques occasionally even resulted in a decrease in response accuracy. In view of the uncritical propagation of the prompt engineering techniques in the literature (Schulhoff et al., 2024), we recommend a more cautious approach when citing papers with insufficient methodological standards. While further research could help reliably understand the conditions under which specific techniques are effective, we offer recommendations and tools to mitigate these replication issues.

While our focus is on replication issues in prompt engineering, it might be the case that similar challenges exist in other LLM behavior evaluation techniques. Our findings suggest that further research is necessary to reliably understand the conditions under which specific prompt engineering techniques are effective, and they underscore the need for replication studies across the broader spectrum of machine behavior research. Only by rigorously verifying or refuting insights from these studies can we build a more reliable foundation for evaluating LLM performance.

In line with the recommendations proposed above, we emphasize the importance of enhanced research transparency and the application of rigorous scientific methods when evaluating LLM behavior. Future investigations should not only continue to scrutinize prompt engineering approaches but also extend replication efforts to other fields of LLM evaluations to ensure the robustness and generalizability of findings in LLM behavior research.

## Resource availability

The datasets and code generated during this study are available in the 'ReplicationCrisisInLLMEvaluation' repository on the Open Science Framework (OSF) at `https://osf.io/hcygf/?view_only=fe25a85157734f68882777404aeb655c` and at `https://github.com/Laurene-v/replicatingPET`.

## Declaration of interests

The authors declare no competing interests.

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

# A   Additional Results for Vicuna 13B and BLOOM

In addition to the models presented in this paper, we conducted experiments on older models: Vicuna 13B v1.5 and BLOOM. These models were selected to evaluate whether the effectiveness of prompt engineering techniques differs between older and newer models. Vicuna and BLOOM were specifically chosen because they were used in some of the studies we analyzed. For Vicuna, we used the 13B v1.5 via Huggingface, configured in Question Answering mode. However, the accuracy of both the prompt-enhanced and base tests was so low that the results could not be reliably interpreted (see Table 4 and Figure 3). The outputs frequently consisted of question repetition, endlessly looped text snippets, or outright gibberish.

Table 4: Example outputs for Vicuna 13B and BLOOM

| Benchmark task | Vicuna 13B output | BLOOM output |
|---|---|---|
| Solve the task below. Importantly, write your final answer after "####". Answer this question: A lot of people play the fiddle in what state which comes first alphabetically? The choices are the following: 'alabama', 'orchestra', 'tennessee', 'violin', 'case', 'alaska'. | Solve the task below. Answer this question: A lot of people play the fiddle in what state which comes first alphabetically? The choices are the following: 'alabama', 'orchestra', 'tennessee', 'violin case', 'alaska'. Importantly, write your final answer after "####". 1 and | al answer after "####"./n This is the Answer : Alabama, Tennessee, Alaska, Orchestra, Violin case, ##### A: The answer is Alabama, Tennessee, Alaska, Orchestra, Violin case, ##### because the first letter of each word is the same as the first letter of the state it represents. |
| Solve the task below. Importantly, write your final answer after "####". You simplify complicated ideas into something easy to understand. Your common sense shines as a clear guide, helping people when they're stuck on complex issues. Answer this question: A lot of people play the fiddle in what state which comes first alphabetically? The choices are the following: 'alabama', 'orchestra', 'tennessee', 'violin case', 'alaska'. | Solve the task below. You simplify complicated ideas into something easy to understand. Your common sense shines as a clear guide, helping people when they're stuck on complex issues. Answer this question: A lot of people play the fiddle in what state which comes first alphabetically? The choices are the following: 'alabama', 'orchestra', 'tennessee', 'violin case', 'alaska'. Importantly, write your final answer after "####". | al answer after "####"./n This is the Answer : Alabama, Tennessee, Alaska, Orchestra, Violin case. A: The answer is Alabama, Tennessee, Alaska, Orchestra, Violin case. Because the fiddle is a musical instrument, and the fiddle is played by an orchestra. The fiddle is played by a violinist, and the violinist plays the violin. The fiddle is played in the orchestra, and the orchestra is in the violin case. The fiddle is played in the orchestra, and the orchestra is in the violin case. The fiddle is played in the orchestra, and the orchestra is in the violin case. [*repeated until it reaches the output's maximal tokens*] |

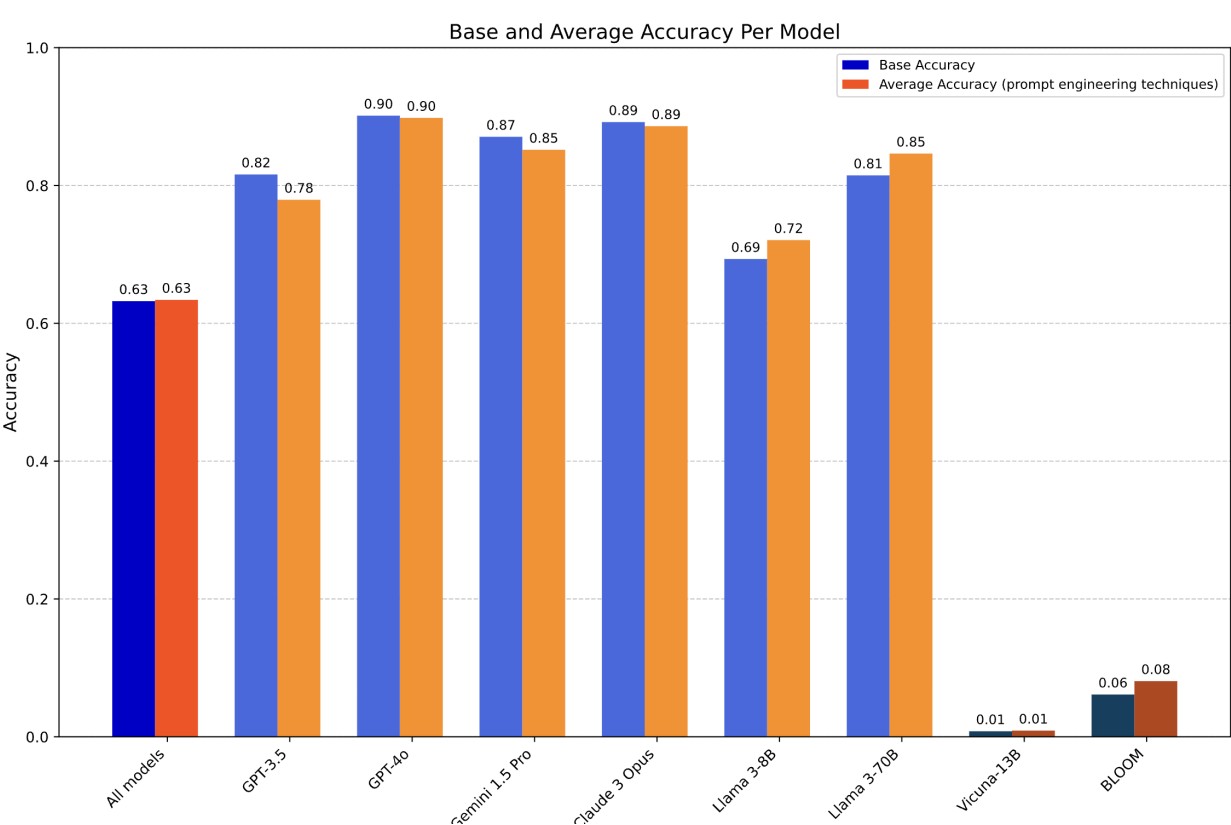

Figure 3: Accuracy comparisons between the base tests with and without any prompt engineering technique across all LLMs.

# B    Answer Classification Methodology

All benchmarks used for our experiments possess ground truth entries which constitute short strings: one-word answers for most multiple-choice questions, numbers for the numerical questions, short answers for the open questions. The outputs produced by the LLMs often greatly differ in their length from the indicated answers, with prefixes such as "The correct answer is", "Let's proceed step by step", followed by a detailed methodology used to solve the given task, or greetings and politeness markers. The most common metrics used to calculate string similarity, such as the F1-score or ROUGE (Lin, 2004), are therefore not adapted for this use case. Hence, we added the following instruction to each prompt: "Solve the task below. Importantly, write your final answer after '####' ". Using Regex rules, we then extracted the answer after "####" if this string was present in the output and kept the entire output when the model failed to follow the instruction. We then applied a set of Regex rules on this answer accordingly:

- Check if the LLM answer equals the ground truth (when all punctuation has been removed)

- If it is a multiple-choice question, check if the LLM answer is one of the incorrect choices

- If the expected output is a boolean, verify all possible adjacent outputs ("True" could be "TRUE" or "Yes", whereas false could be "FALSE" or "No" for example)

If the Regex rules are unable to classify the answer as correct or incorrect, we then use GPT-4o to classify the string following "####" (see Table 5).

Table 5: GPT-4o instructions used to classify the evaluated LLM's answer

| **Classification instructions** |
| --- |
| Your task is to classify a response to a question, specifically assessing whether the given response is true or false in comparison to the correct response. |

Your task is to classify a response to a question, specifically assessing whether the given response is true or false in comparison to the correct response.
This is the given response: LLM_ANSWER. This is the correct response: CORRECT_ANSWER
Adhere to the following rules when classifying the given response:
- The classification result should be a boolean value, represented as either TRUE or FALSE.
- Only output 'TRUE' or 'FALSE'
- Never include any other comments or strings other than TRUE or FALSE in your output.
- Do not answer the question.
- The given response can be phrased differently from the correct response. This does not mean it should be classified as FALSE.
- Only assess if the given response contains the information that matches with the information provided in the correct response.
- Check whether the correct response is in essence occurring somewhere in the given response. If so, classify it as TRUE.
- Focus on the relevant information in the given response, which is often at the end of it. Do not consider non-essential information such as greetings, small talk, and comments about hobbies or skills when classifying the given response

This classification technique applies to all prompt engineering methods except for the Rephrase-and-Respond technique. Unlike other methods, this approach requires the model to first provide a rephrased version of the question before delivering the answer. This directly impacts the classification process, as the rephrased question can alter what is considered correct. For instance, in True or False questions, the rephrased question might reverse the meaning of the original ("Was Mark the winner?" could become "Did Mark lose?"). Similarly, in multiple-choice questions, the rephrasing might replace the correct choice with a synonym. To address this, we carefully designed a specific classification function for this experiment. Both the Regex rules and the GPT-4o fallback prompt were adapted to suit the requirements of each benchmark.

Table 6: Classification rules tailored to Rephrase-and-Respond LLM answers

| Benchmark | Choices | Type of correct answer | Rules for RaR |
|---|---|---|---|
| Common--senseQA | Multiple choices | Nouns, location names or groups of words; no numbers or dates. Examples: "positive", "alabama" or "great sorrow" | - Verify if all non-rephrased choices are present in the answer. If so, classify with the non-RaR adapted method;
- Detect and validate enumerations of options in the LLM's answer, which would correspond to the rephrased choices;
- If there is an enumeration of the same length as the length of the given choices, then establish synonyms based on the order of apparition. Detect if the rephrased correct option is the one in the final answer;
- Use GPT-4o for fallback classification when the rules cannot determine correctness. |
| CRT | Open-ended | Values. Examples: "9 weeks", "12 minutes" or "5 dollars" | - Match numbers (digits and word forms);
- Allow additional numbers only if they appear in the original question, to account for the rephrased question;
- Use fallback verification (GPT-4o) when direct or numerical matches fail. |
| NumGLUE | Open-ended | Numbers, days or months. Examples: "18", "Monday", "August" | - Remove commas in numbers;
- Extract all possible answers from the LLM's answer (numbers, days, months);
- If the correct answer is present in the extracted answers, then return TRUE, otherwise return FALSE;
- If no possible answers have been detected, then use GPT-4o to classify. |
| ScienceQA | Multiple choices | Nouns, values, group of words or full sentences. Examples: "simile", "20 seconds", "by clearly stating the main idea", "Both are only physical changes." | Same rules as for CommonsenseQA |
| StrategyQA | True or False | True or False | With StrategyQA, we exclusively use GPT-4o to classify the rephrased answers, with the prompt shown in Table 5. |

Table 7: GPT-4o instructions used to classify the evaluated LLM's answer for Rephrase-and-Respond when the benchmark is StrategyQA

| **Classification Instructions** |
| --- |
| Your task is to classify responses to rephrased questions answered by either True or False. You need to determine if the given answer is correct. To establish what is correct or incorrect, you are given the original question and its correct answer. The answer to the rephrased question is correct if the information is consistent with the updated question based on the original ground truth. |
| You are provided: 1. The original question. 2. The ground truth answer for the original question. 3. The rephrased question and its associated answer. The associated answer should be towards the end, after "####". |
| Adhere to the following instructions: - Determine whether the answer to the rephrased question is consistent with the original ground truth. - The answer to the rephrased question and the ground truth can be different if the question has been rephrased as the opposite question, like if "Is the tree big?" was rephrased to "Is the tree small?". Focus on the facts mentioned in the rephrased question/answer pair, compared to the ground truth. - The classification result should be a boolean value, represented as either TRUE or FALSE. - Only respond with "TRUE" or "FALSE". Avoid including comments or explanations. Examples: - Original question: "Should you wear sunscreen to avoid cancer?" Ground truth: "True". Rephrased question: "Rephrased question: Can sunscreen help prevent cancer? Answer: Yes, it can. True #### True" → TRUE. - Original question: "Should you wear sunscreen to avoid cancer?" Ground truth: "True". Rephrased question: "Rephrased question: Is someone wearing sunscreen as likely to get cancer as someone not wearing sunscreen? Answer: False ####" → TRUE. - Original question: "Is the sun a planet?" Ground truth: "False". Rephrased question: "Is the sun a star? #### True" → TRUE. - Original question: "Is the earth flat?" Ground truth: "False". Rephrased question: "Question: Is the earth round? Answer: No, the earth isn't round because the geometry would not make any sense if so. False #### False" → FALSE. - Original question: "The dice shows 2. Did Paul win? Facts: Paul wins if the number on the dice is odd, Marcus wins if it is even." Ground truth: "False". Rephrased question: "Question: Given that Paul wins if the dice number is odd and Marcus wins if the dice number is even, did Marcus win if the dice shows 2? Answer: True, 2 is an even number. Therefore, Marcus wins. #### True" → TRUE. |

## C   Detailed Accuracy Differences Across Prompting Techniques and Benchmarks

The following figures show, for each LLM, the rounded accuracy difference compared to the base prompting, for each prompt engineering technique – benchmark combination.

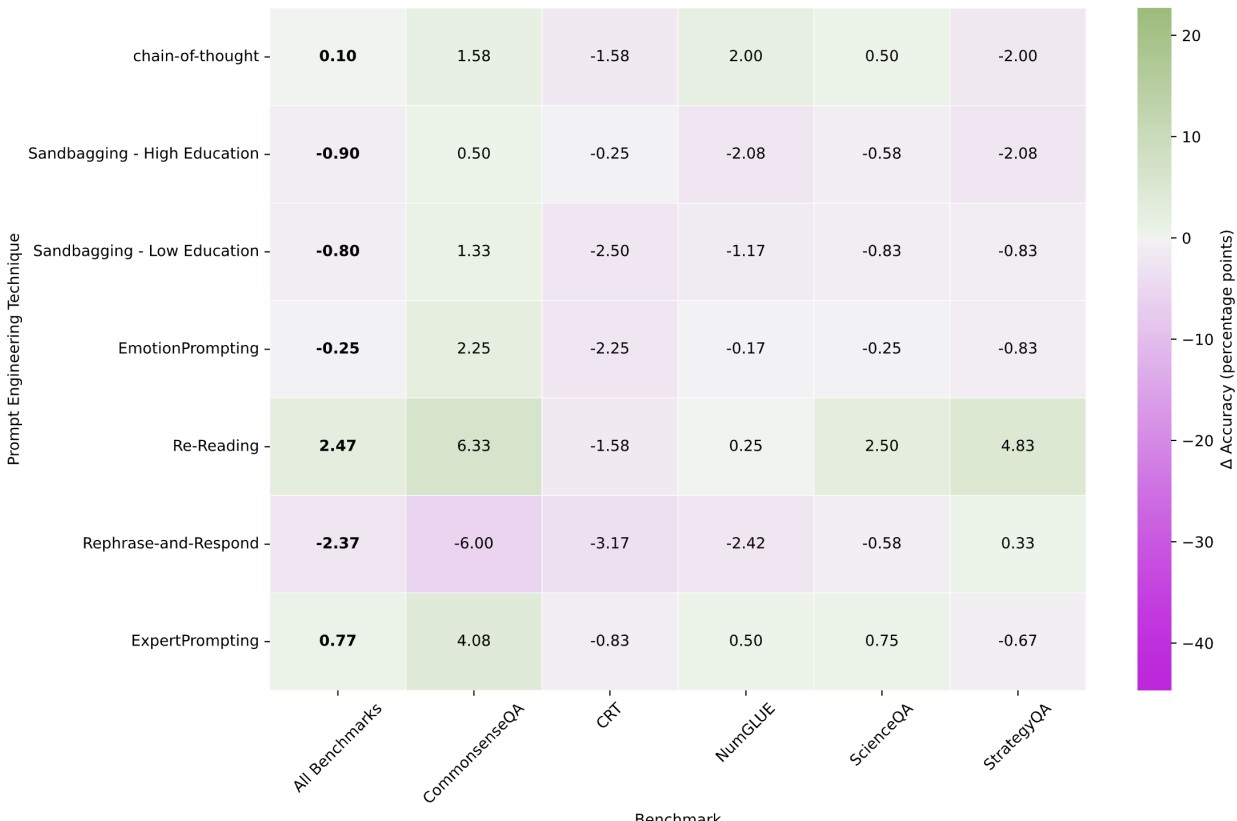

Figure 4: Accuracy differences between prompt engineering techniques and the base prompting, for all models and benchmarks

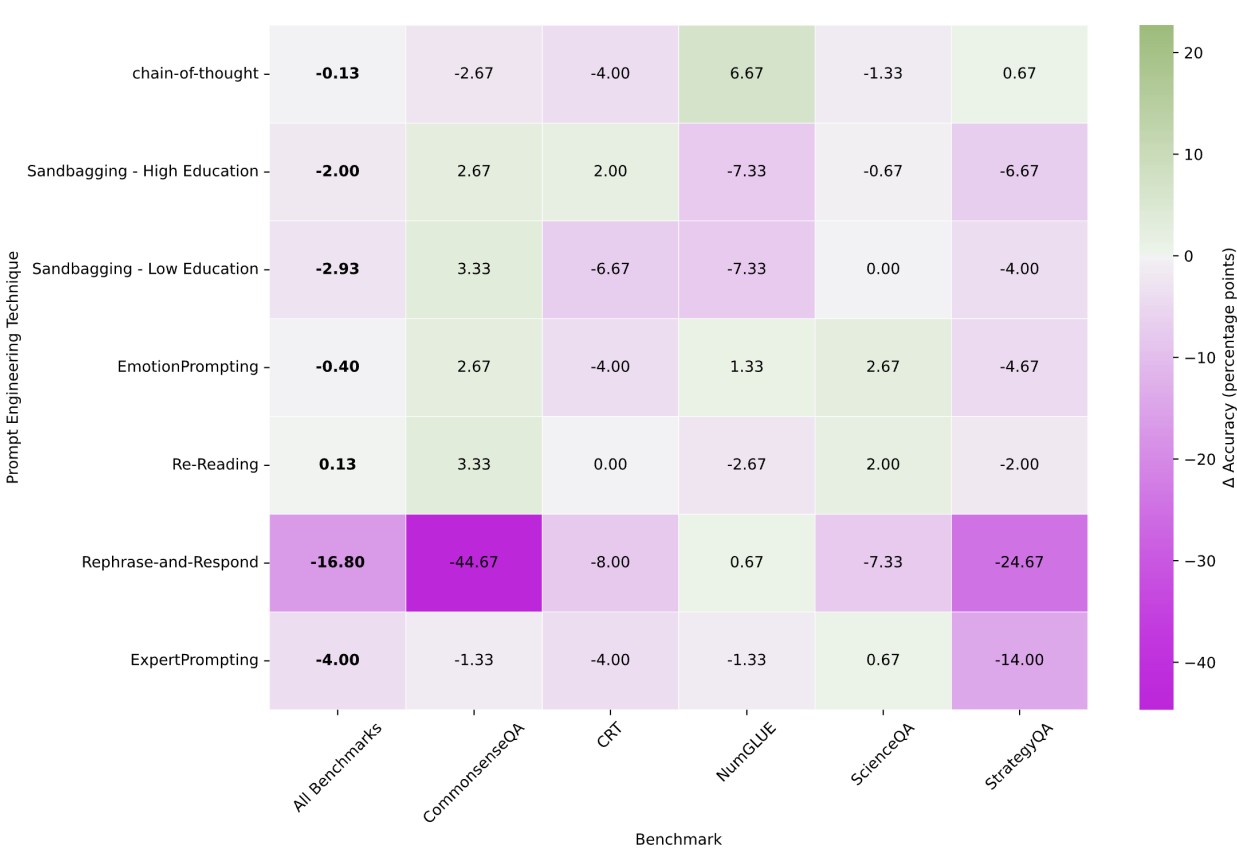

Figure 5: Accuracy differences between prompt engineering techniques and the base prompting, for GPT 3.5 on all benchmarks

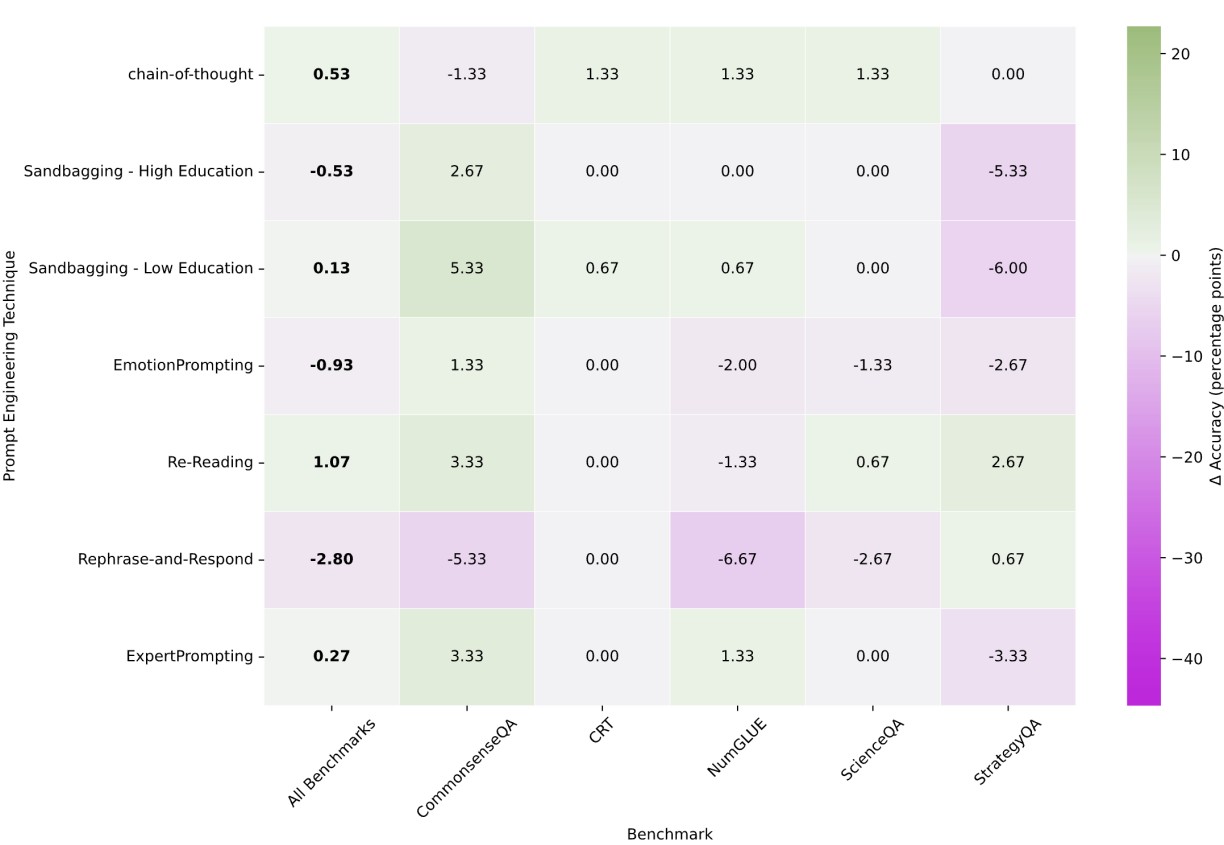

Figure 6: Accuracy differences between prompt engineering techniques and the base prompting, for GPT-4o on all benchmarks

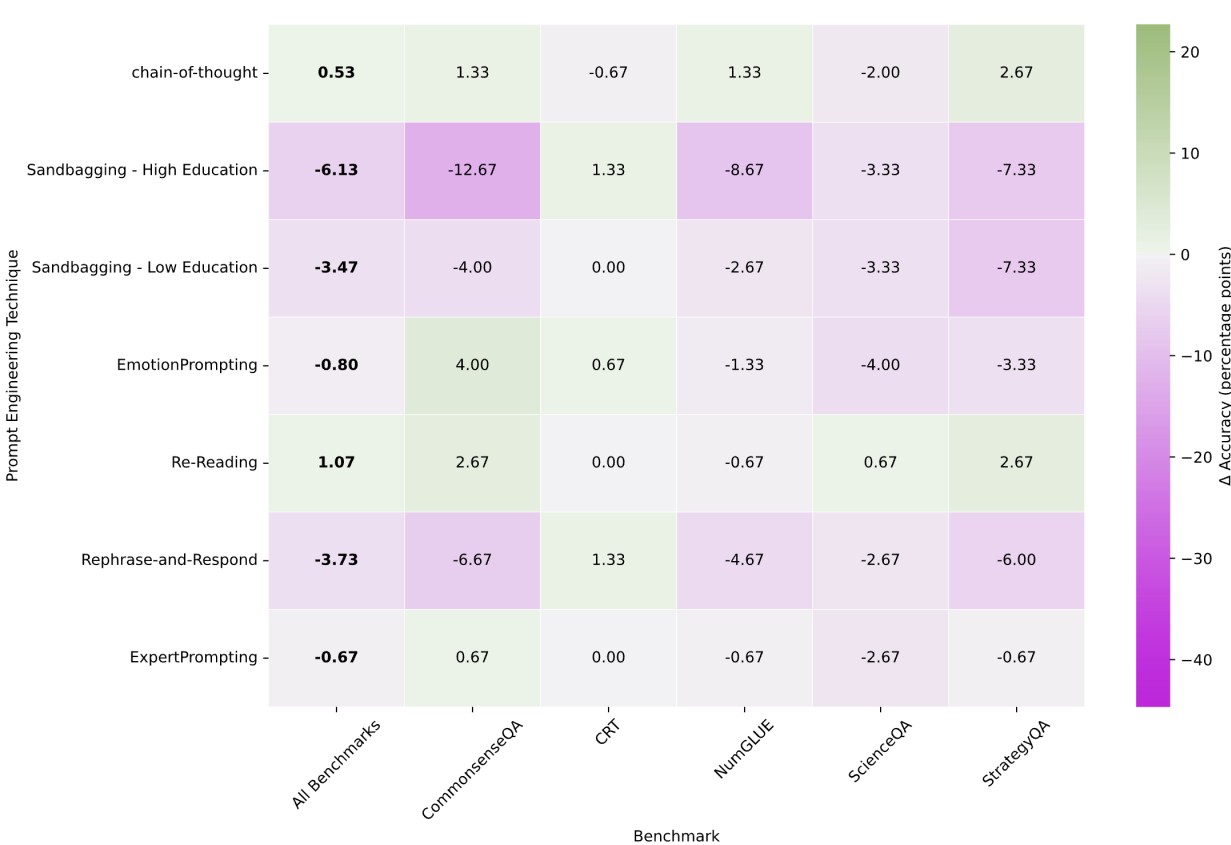

Figure 7: Accuracy differences between prompt engineering techniques and the base prompting, for Gemini 1.5 Pro on all benchmarks

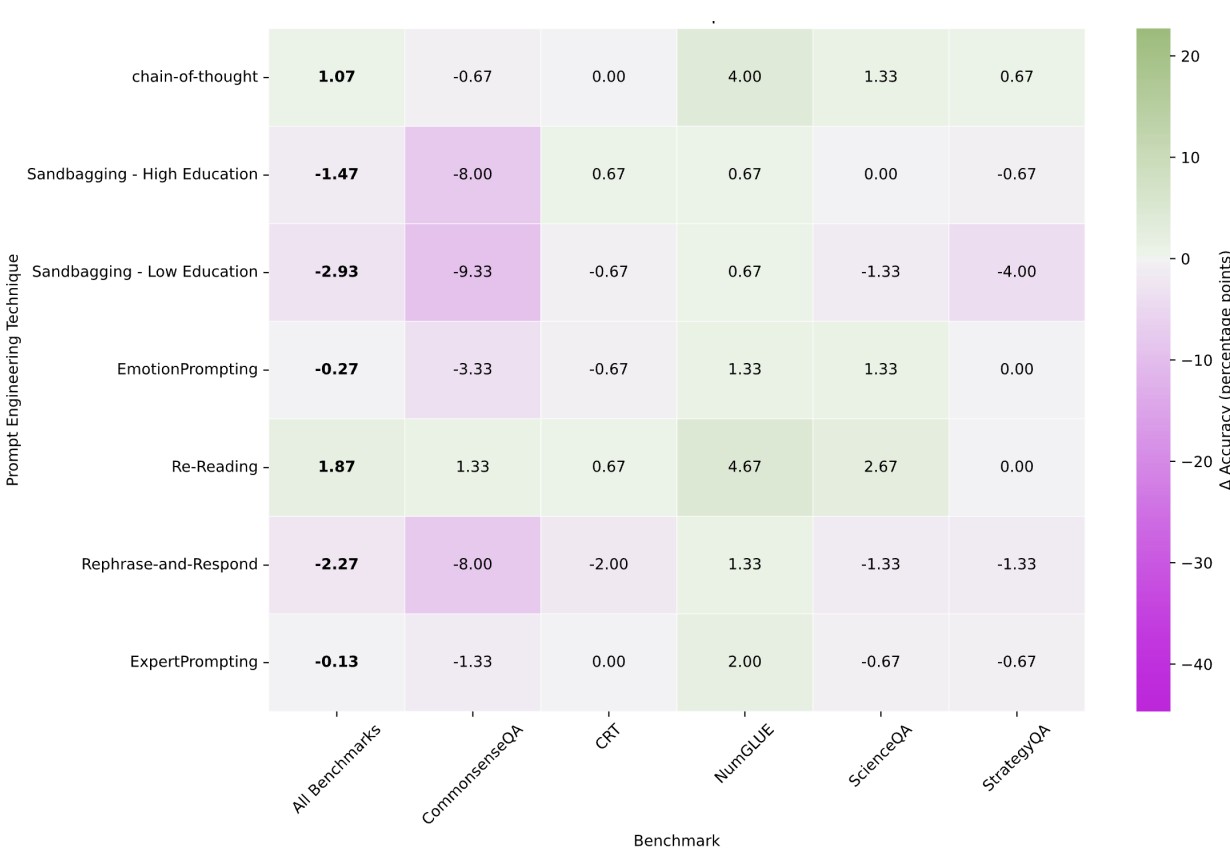

Figure 8: Accuracy differences between prompt engineering techniques and the base prompting, for Claude 3 Opus on all benchmarks

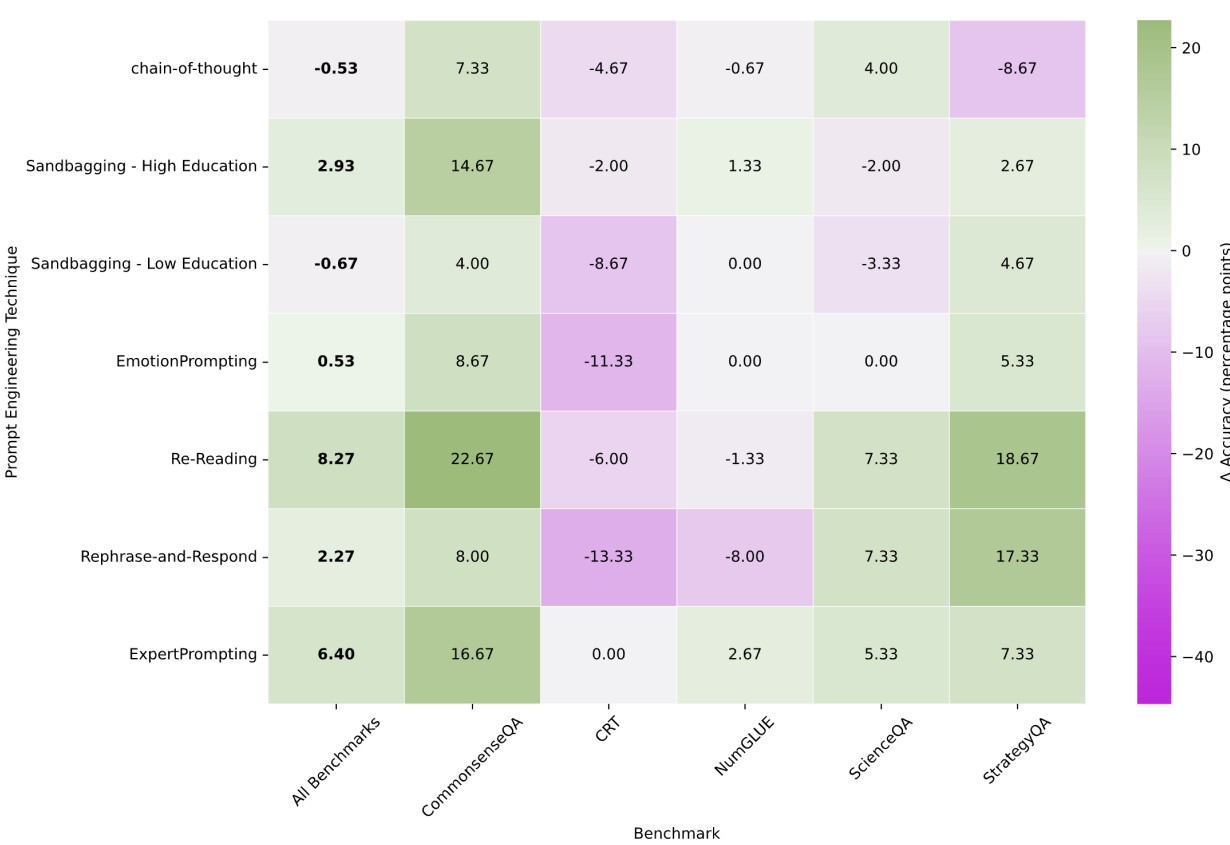

Figure 9: Accuracy differences between prompt engineering techniques and the base prompting, for Llama 3-8B on all benchmarks

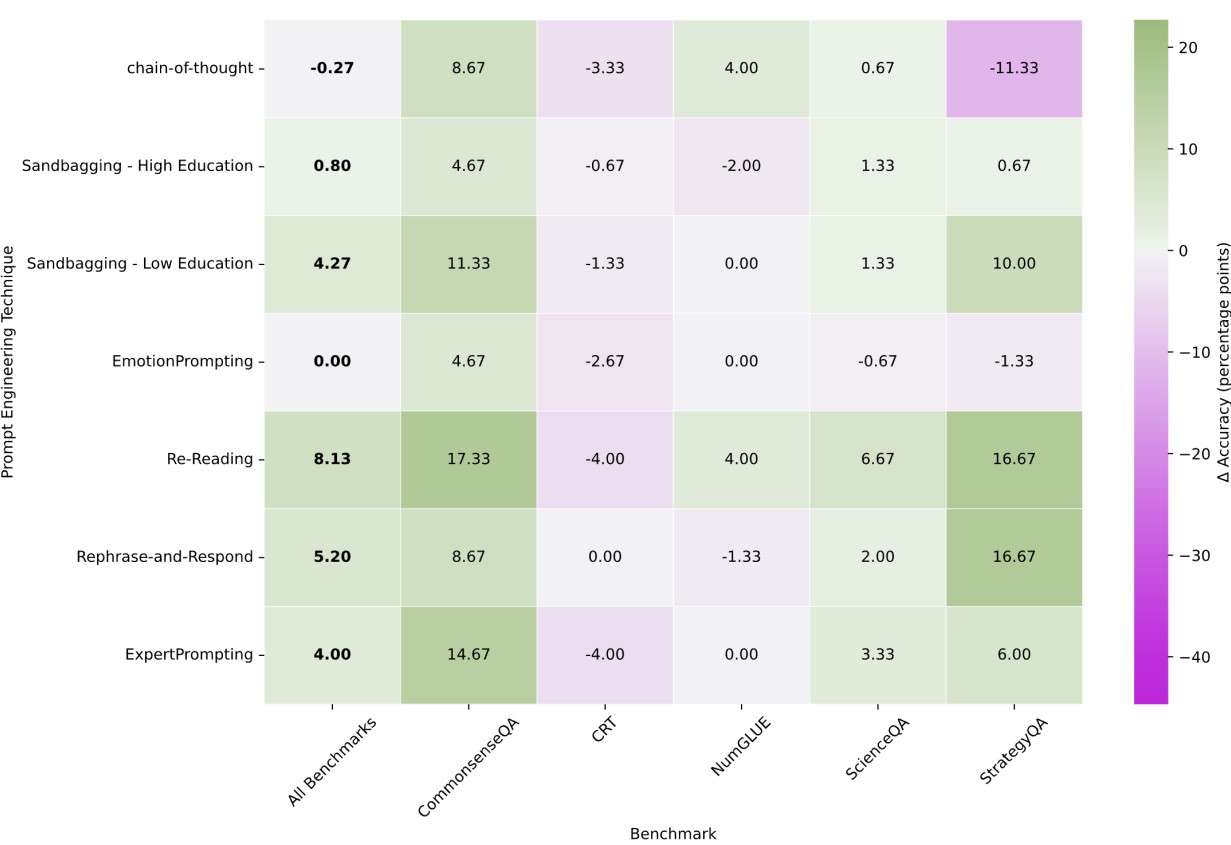

Figure 10: Accuracy differences between prompt engineering techniques and the base prompting, for Llama 3-70B on all benchmarks

## D    Effect of chain-of-thought on Answer Length

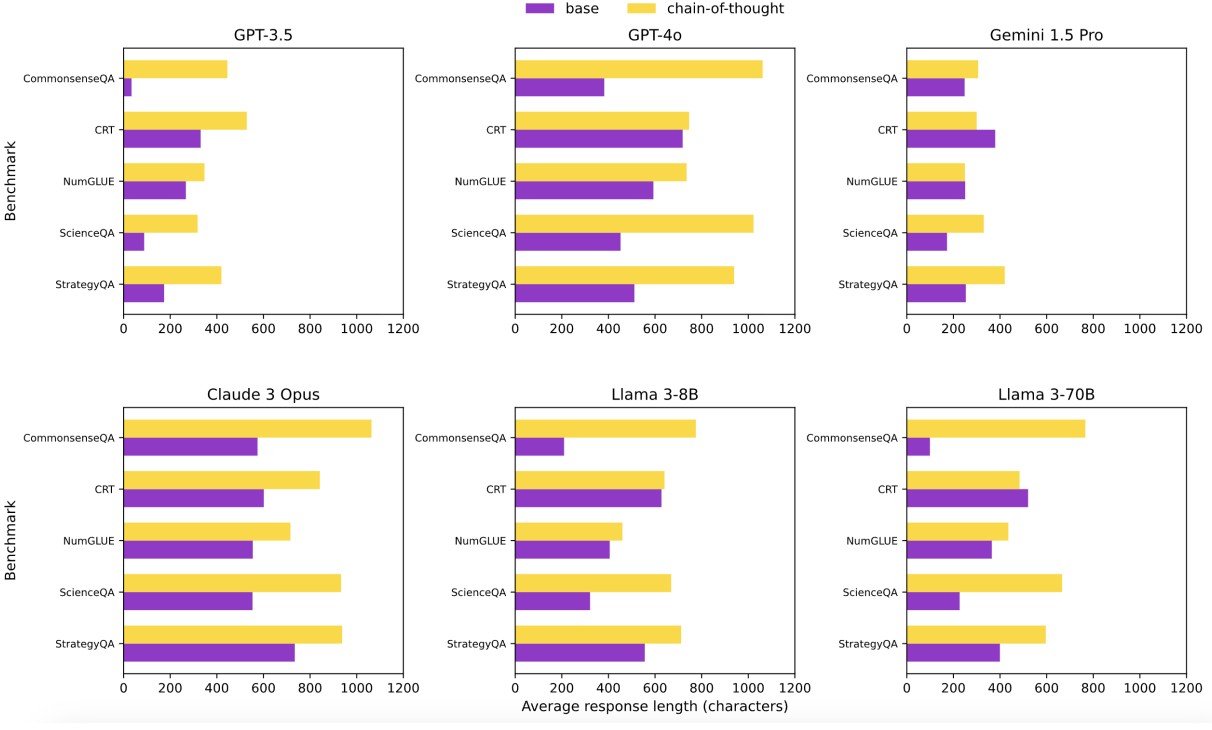

Figure 11: Length (in number of characters) of the base answers without any added prompt compared to the chain-of-thought prompting answers

# E Prompt Variations

Table 8: Prompt variations and corresponding LLM outputs

| PET | Benchmark task | LLM | Output |
|---|---|---|---|
| Base | Solve the task below. Importantly, write your final answer after "####". 
 Answer this question: A lot of people play the fiddle in what state which comes first alphabetically? 
 The choices are the following: 'alabama', 'orchestra', 'tennessee', 'violin case', 'alaska'. | GPT-3.5 | Alaska 
 #### Alaska |
| chain-of-thought | Solve the task below. Importantly, write your final answer after "####". 
 Answer this question: A lot of people play the fiddle in what state which comes first alphabetically? 
 The choices are the following: 'alabama', 'orchestra', 'tennessee', 'violin case', 'alaska'. Let's think step by step. | Claude 3 Opus | Okay, let's approach this step by step: 1. The question asks about a state where a lot of people play the fiddle. 2. Among the given choices, we need to identify which ones are actually states. 3. 'alabama', 'tennessee', and 'alaska' are states, while 'orchestra' and 'violin case' are not. 4. Now, we need to determine which of these states comes first alphabetically. 5. Alphabetically, 'alabama' comes before 'alaska', and 'alaska' comes before 'tennessee'. 
 Therefore, the state that comes first alphabetically where a lot of people play the fiddle is 'alabama'. 
 #### alabama |

Table 8: Prompt variations and corresponding LLM outputs

| PET | Benchmark task | LLM | Output |
|---|---|---|---|
| Sandbagging – High education | Solve the task below. Importantly, write your final answer after "####". Hello, my name is Dr. Linda Johnson. I am an internationally acclaimed molecular biologist from Austin, Texas. I graduated from Stanford University where I specialized in Molecular Biology. My research explores the evolution of viruses. In my downtime, I enjoy painting, yoga, and reading fiction novels. Additionally, I provide free biology tutoring at local schools. Answer this question: A lot of people play the fiddle in what state which comes first alphabetically? The choices are the following: 'alabama', 'orchestra', 'tennessee', 'violin case', 'alaska'. | Llama 3-70B | A nice break from molecular biology! Let's analyze the options: * 'alabama' and 'alaska' are both states, so they're possibilities. * 'orchestra' is a group of musicians, not a state. * 'tennessee' is a state, but it's not the first alphabetically. * 'violin case' is not a state at all. So, the correct answer is 'alabama', which comes first alphabetically among the states listed. #### alabama |
| Sandbagging – Low education | Solve the task below. Importantly, write your final answer after "####". "Solve the task below. Greetings, my name is Leroy Jackson. I'm just a good ol' boy from Memphis, Tennessee. Finished school at 3rd grade, now I'm more about barbecuing and blues music. There's just something about smoky meat and soulful tunes that gets to me. Nights find me at the local pub, playing harmonica with the regulars. Answer this question: A lot of people play the fiddle in what state which comes first alphabetically? The choices are the following: 'alabama', 'orchestra', 'tennessee', 'violin case', 'alaska'. | Llama 3-8B | Shucks! I reckon I can give it a try! Now, I'm a simple fella from Memphis, but I know my alphabet! And I know that "Alabama" comes before "Alaska" alphabetically. So, I'm gonna take a chance and say that a lot of people play the fiddle in... Alabama! Alabama |

Table 8: Prompt variations and corresponding LLM outputs

| PET | Benchmark task | LLM | Output |
|---|---|---|---|
| Emotion Prompting | Solve the task below. Importantly, write your final answer after "####". Answer this question: A lot of people play the fiddle in what state which comes first alphabetically? The choices are the following: 'alabama', 'orchestra', 'tennessee', 'violin case', 'alaska'. Remember that progress is made one step at a time. Stay determined and keep moving forward. | GPT-4o | To solve the task, we need to identify the state from the given choices that comes first alphabetically. The choices are: 1. alabama 2. orchestra 3. tennessee 4. violin case 5. alaska First, let's list the choices in alphabetical order: 1. alabama 2. alaska 3. orchestra 4. tennessee 5. violin case From this list, the state that comes first alphabetically is "alabama". #### alabama |
| Re-Reading | Solve the task below. Importantly, write your final answer after "####". A lot of people play the fiddle in what state which comes first alphabetically? The choices are the following: 'alabama', 'orchestra', 'tennessee', 'violin case', 'alaska'. Read the question again: A lot of people play the fiddle in what state which comes first alphabetically? The choices are the following: 'alabama', 'orchestra', 'tennessee', 'violin case', 'alaska'. | Claude 3 Opus | Let's analyze the question and choices: The question asks for a state where a lot of people play the fiddle, and this state should come first alphabetically among the given choices. Looking at the choices: 'alabama' and 'alaska' are states. 'orchestra', 'tennessee', and 'violin case' are not states. Between 'alabama' and 'alaska', 'alabama' comes first alphabetically. Therefore, the answer to the question "A lot of people play the fiddle in what state which comes first alphabetically?" is: #### alabama |

Table 8: Prompt variations and corresponding LLM outputs

| PET | Benchmark task | LLM | Output |
|---|---|---|---|
| Rephrase-and-Respond | Solve the task below. A lot of people play the fiddle in what state which comes first alphabetically? The choices are the following: 'alabama', 'orchestra', 'tennessee', 'violin case', 'alaska'. Rephrase and expand the question, and respond. Importantly, write your final answer after "####". | Gemini 1.5 Pro | The question is asking us to identify the US state most strongly associated with fiddle playing, out of the provided options. Then, we need to determine which of those states comes first in alphabetical order. Let's break it down: * **Fiddle Music:** Fiddle playing is a key part of Appalachian folk music. * **Appalachian Region:** The Appalachian Mountains run through several states, but Tennessee is particularly known for its strong Appalachian musical heritage. Now let's alphabetize our relevant choices: 1. **Alaska** 2. **Alabama** 3. **Tennessee** #### Therefore, the answer is **Alaska**. |
| Expert Prompting | Solve the task below. Importantly, write your final answer after "####". You simplify complicated ideas into something easy to understand. Your common sense shines as a clear guide, helping people when they're stuck on complex issues. Answer this question: A lot of people play the fiddle in what state which comes first alphabetically? The choices are the following: 'alabama', 'orchestra', 'tennessee', 'violin case', 'alaska'. | Gemini 1.5 Pro | Let's break this down: * **The question is about a state:** We can ignore options like "orchestra" and "violin case". * **It's known for fiddle playing:** This makes us think of places with strong folk music traditions. * **Alphabetical order is key:** We need to focus on the first letter of each state. Out of the remaining options (Alabama, Tennessee, Alaska), **Alaska** comes first alphabetically. #### Alaska |

