# OpenReview forum: "Prompt Engineering Techniques for Language Model Reasoning Lack Replicability"
_TMLR — Accepted by TMLR_

### Review · Reviewer_VSak · 2025-09-13

**Summary Of Contributions:**

The paper evaluates whether popular prompt engineering techniques (Zero-shot CoT Prompting, Sandbagging, EmotionPrompting, Re-Reading, Rephrase-and-Respond, ExpertPrompting) actually improve reasoning accuracy across several LLMs (GPT-3.5, GPT-4o, Claude 3 Opus, Gemini 1.5 Pro, Llama-3 8B/70B; Vicuna/BLOOM) on five reasoning benchmarks (CommonsenseQA, StrategyQA, NumGLUE, ScienceQA, CRT).
Surprisingly, compared to prior works, the authors report mostly no statistically significant gains, except partial, model-specific improvements (e.g., Re-Reading and Rephrase-and-Respond on Llama-3 for StrategyQA).
The authors argue prior positive claims often rely on method choices (quality vs accuracy, cherry-picked prompts) and propose practical recommendations to strengthen replicability and statistical rigor in future prompt engineering research.

**Additional Comments:**

N/A

**Audience:**

Yes

**Audience Explanation:**

Some TMLR readers will be interested in this paper, because it challenges a prevailing narrative: popular prompt-engineering techniques reliably boost accuracy.
Showing largely null effects across multiple models and benchmarks is intrinsically newsworthy for practitioners.

**Broader Impact Concerns:**

The reviewer does not identify any ethical concerns.

**Claims And Evidence:**

Yes

**Claims Explanation:**

The paper's claim, popular prompt engineering techniques often fail to deliver accuracy gains, is supported by careful analysis.
The authors test six techniques across six models on five benchmarks.
They assess outcomes using chi-squared tests and report p-values.

**Requested Changes:**

My main concern is that the paper’s depth feels limited.
The main message of the paper is “popular prompt engineering techniques do not help on various models and datasets”, without explaining why its results differ from prior reports.
As it stands, this single message feels insufficient for acceptance.
The contribution would be much stronger if the authors provide following two analyses:

- Reproduce the original setups: Attempt to match the setups from the prior papers (same prompts, metrics, model versions, etc) to verify whether earlier gains can be recovered.

- Diagnose the divergence: If those gains are reproducible in the original setups, analyze why this paper’s configuration yields different outcomes.

To deepen the work, please also discuss plausible mechanisms behind this divergence.
Two illustrative possibilities are:

(a) Capability maturation: earlier pretrained LLMs may have lacked intrinsic reasoning ability, so prompt engineering exposed latent capacity; newer models may already leverage these behaviors, reducing the marginal benefit.

(b) Provider-side prompting: modern APIs often include sophisticated system prompts (sometimes akin to instruction-following or CoT), leaving less headroom for user-level prompt tweaks.

These two are just possibilities, and there may be additional contributing factors.
What matters is identifying (and, where possible, validating) at least one concrete mechanism that explains why your results diverge from prior reports.
This would meaningfully deepen the paper.

Also, the models used in the paper are somewhat outdated as of September 2025.
Adding an evaluation with more recent LLMs would strengthen the paper.

Lastly, here are some minor mistakes.

- Correct appendix references (e.g., "Appendix 5" $\to$ "Appendix E")

- In the discussion section, "suggest that further research is necessary ..." appears twice.

---

### Review · Reviewer_sKs3 · 2025-09-14

**Summary Of Contributions:**

The paper highlights the issue of replicability in zero-shot prompt engineering techniques for Language Model Reasoning. They state that clear methodological guidelines for evaluating these techniques are lacking. The issue pointed out in the paper is crucial and of great importance to the field. In the paper, authors compare different zero-shot prompting techniques (like chain-of-thought, emotional prompting, expert prompting, etc.) on different LLMs like GPT-3.5, GPT-4o, Llama 3, etc. They created a subset of datasets (manually checked) from different reasoning benchmarks and tested different prompting techniques on different LLMs. Finally, authors also propose recommendations based on the problems of previous evaluations of prompting strategies.

## Strengths -
- The paper raises the question of replicability which most prompt based papers in LLMs lack, they highlight the probable problems and propose recommendations for better evaluations strategies.
- The paper highlights some key problems with the evaluation of current zero-shot methods like the use of cherry picked prompts in Emotion Prompting.
- Good recommendations are provided by the authors which is backed by thorough overview of previous prompting methods. They highlight the drawbacks of some zero shot prompt papers in their evaluations and suggest some useful recommendations.

## Weaknesses -
I have put the weakness in requested changes section.

**Audience:**

Yes

**Audience Explanation:**

Yes the problem is genuine and there is a need of better evaluation criteria, lot's of people will be interested in it.

**Claims And Evidence:**

No

**Claims Explanation:**

I am not convinced about the claims of the paper, I have put my exact questions in requested changes and will be interested in authors answers.

**Requested Changes:**

## Weakness -

I have highlighted the critical points necessary to get my recommendation by [critical] at the start of the points.

I see two major weaknesses in the paper, first one is on the reliability of the scores (it could be because LLMs already use these prompting methods internally or problems in benchmark datasets) and the second one is the missing explanation of why those prompts do work in other papers. I will expand these two points and also put 1-2 minor points.

- [Critical]: For reliability, one of my major concern is the limited dataset size used in the paper. For example in zero shot prompting the original paper used 12 different benchmarks and similarly for Rephrase and Respond CommonsenseQA dataset used in the paper had 12,102 data points. These dataset/benchmarks are much large compared to the limited data points picked in this paper, even though they are manually double-checked, according to me they are few in number to say something concrete.
- [Critical]: For reliability, as pointed out by the authors closed source LLMs might be using these prompting techniques internally so why would one use those models for comparison? Even in [1] they point out something similar (for GPT-4o). In addition to this, in appendix Llama 3 (8B and 70B) shows some improvement even for 95% confidence, which should be considerable.
- [Critical]: For reliability. Currently the data points are drawn randomly from the dataset, could there be a better way to pick ‘good’ data points from these datasets such that the sample still maintains some properties (like complexity of the questions), this would give better insights without drastically changing the datasets.
- [Critical]: For missing explanations. I feel the reason pointed for inconsistency in the results of prompting is still limited and very general. I understand there could be a role of bad data, specific experimental setting or others but some more finer reasoning or explanation is required. Because of this missing analysis the question of enough novelty in the paper is also raised.
    - To give an example of more finer explanations, one could look into which points in the datasets (entire dataset with ‘bad’ points) actually worked better and whether ‘bad’ points were actually responsible for the increase of scores for these prompting methods.
- [Critical]: I have also mentioned a paper [1] which looks very relevant, I would request the authors to compare and contrast their work with the paper and cite it in their paper.
- In my opinion the paper is predominantly text-heavy, particularly in Section 3. The inclusion of additional tables and figures would significantly improve readability and help readers better understand it.

## References
[1]
@article{DBLP:journals/corr/abs-2411-02093,
  publtype={informal},
  author={Guoqing Wang and Zeyu Sun and Zhihao Gong and Sixiang Ye and Yizhou Chen and Yifan Zhao and Qingyuan Liang and Dan Hao},
  title={Do Advanced Language Models Eliminate the Need for Prompt Engineering in Software Engineering?},
  year={2024},
  cdate={1704067200000},
  journal={CoRR},
  volume={abs/2411.02093},
  url={https://doi.org/10.48550/arXiv.2411.02093}
}

---

### Review · Reviewer_as4w · 2025-09-28

**Summary Of Contributions:**

The paper presents an extensive replication to evaluate the performance of LLMs in several benchmarks. The empirical obtained from the study shows a significant deviation from the reported results in the literature. Specifically, in the *chain-of-thoughts promting*, the replication from the paper shows no statistical difference compared to the improvement reported in previous studies. Similar observations (e.g., no significant or only a slight improvement) are also in * Sandbagging*, *EmotionPrompting*, *Re-Reading*, * Rephrase-and-Respond* and * ExpertPrompting*. Such results have, therefore, motivated the authors to make several recommendations when evaluating LLMs, including benchmarks, models, model versions (or model updates) and the outputs.

**Additional Comments:**

**Minor errors/comments**
- The double quotes should be written as ``some quote'', not the one copy from other text editor as the one seen in subsection 2.1 at the bullet point "EmotionPrompting".
- The citation in subsection 3.6:  B. Xu et al. (21) is inconsistent with the remaining citations in the paper.

**Audience:**

Yes

**Audience Explanation:**

At the current form, the paper, to me, is another reproducible effort to verify previous claims, rather than contributing further to the research in this subfield. The authors did argue the difference between reproducibility and replication, however, the current study claiming as "replication" does not reveal much insights as well as a solid or a work-around way to fix. The recommendations at the end are, even though theoretically appealing, too vague without specific actions or benchmarks for the community to follow.

**Broader Impact Concerns:**

The paper reports the problem in the current evaluation of LLMs using prompting techniques, and hence, impactful.

**Claims And Evidence:**

No

**Claims Explanation:**

I do appreciate the effort done by the authors to re-evaluate the performance of several LLMs on different benchmarks. Although the paper points out several inconsisent results reported in the literature, such as no improvement but was reported as significantly improved, this is unfortunately a serious-but-known issue in machine learning research in general. This issue has led to the effort of reproducibility without much success, and now, it has been resurfaced as evidenced in the paper.

Nevertheless, the contribution of the paper is still insufficient and simply stops at evaluating on their own benchmarks without further analyzing the causes to fix. For example, in the recommendation for the benchmarks, the paper argues about the quality, such as typos or errors, which contributes to the inadequate benchmarks. However, a controlled study, in which two sets of results: one for the one with typos and errors, and the other for the one after fixing, would give a more solid conclusion.

**Requested Changes:**

The paper would be strong for a publication if the recommendations are more specific, such as a proper benchmarks for the community to use, or a way to fix the output of LLMs (similar to fixing a random seed).

---

### Decision · Action_Editor_hFYA · 2025-11-20

**Recommendation:** Accept as is

**Audience:**

Yes

**Audience Explanation:**

Prompt engineering is a practical research topic. This paper presents some interesting findings and provides meaningful recommendations to the community.

**Claims And Evidence:**

Yes

**Claims Explanation:**

This paper presents a very comprehensive empirical study to evaluate the performance of several prompt engineering techniques across various settings. Results show that the evaluated prompt engineering techniques do not show any statistically significant improvement compared to naive baselines. Reviewers raised some concerns on missing literature, evaluation metrics, paper presentations, etc., which have been well addressed in the revised paper.